# AI-NAOS: An AI-Based Nonspherical Aerosol Optical Scheme for Chemical Weather Model GRAPES_Meso5.1/CUACE

Xuan Wang[1], Lei Bi[1], Hong Wang[2], Yaqiang Wang[2], Wei Han[3], Xueshun Shen[3], Xiaoye Zhang[2]

[1]Key Laboratory of Geoscience Big Data and Deep Resource of Zhejiang Province, School of Earth Sciences, Zhejiang University, Hangzhou, China
[2]Key Laboratory of Atmospheric Chemistry of CMA, Institute of Atmospheric Composition, Chinese Academy of Meteorological Sciences, Beijing, China
[3]CMA Earth System Modeling and Prediction Centre (CEMC), China Meteorological Administration, Beijing, China

*Correspondence to*: Lei Bi (bilei@zju.edu.cn)

**Abstract** The AI-based Nonspherical Aerosol Optical Scheme (AI-NAOS) is a newly developed aerosol optical module that improves the representation of aerosol optical properties for radiative transfer simulations in atmospheric models. It incorporates the nonsphericity and inhomogeneity (NSIH) of internally mixed aerosol particles through a deep learning method. Specifically, the AI-NAOS considers black carbon (BC) as fractal aggregates and models soil dust (SD) as super-spheroids, encapsulated partially or completely with hygroscopic aerosols such as sulfate, nitrate, and aerosol water. To obtain AI-NAOS, a database of the optical properties for the models was constructed using the invariant imbedding T-matrix method (IITM), and deep neural networks (DNN) were trained based on this database. In this study, the AI-NAOS was integrated into the mesoscale version 5.1 of Global/Regional Assimilation and Prediction System with Chinese Unified Atmospheric Chemistry Environment (GRAPES_Meso5.1/CUACE). Real-case simulations were conducted during a winter with high pollution, comparing BC aerosols evaluated using three schemes with spherical aerosol models (external-mixing, core-shell, and volume-mixing) and the AI-NAOS scheme. The results showed that NSIH effect led to a moderate estimation of absorbing aerosol optical depth (AAOD) and obvious changes in aerosol radiative effects, short-wave heating rates, temperature profiles, and boundary layer height. The AAOD values based on three spherical schemes were 70.4%, 125.3%, and 129.3% over Sichuan Basin, benchmarked to the AI-NAOS results. Compared to the external-mixing scheme, the direct radiative effect (DRE) induced by the NSIH effect reached +1.6 W/m$^2$ at the top-of-atmosphere (TOA) and -2.9 W/m$^2$ at surface. The NSIH effect could enhance the short-wave heating rate, reaching 23%. Thus, the warming effect at 700 hPa and the cooling effect on the ground were strengthened by 21% and 13%, reaching +0.04 and –0.10 K, which led to a reduction in the height of the Planetary Boundary Layer (PBL) by –11 meters. In addition, the precipitation was inhibited by the NSIH effect, causing a 15% further decrease. Therefore, the NSIH effects demonstrated their non-negligible impacts and highlighted the importance of incorporating them into chemical weather models.

## 1 Introduction

Aerosols play a significant role in the Earth's climate system through various pathways, including direct scattering and absorbing radiation, and also indirectly, by affecting cloud formation and dimming snow, hence influencing the energy budget and climate processes. However, lager uncertainties regarding aerosol radiative forcing still exist due to challenges in identifying their distribution, transportation, and physical properties (Yu et al., 2006). The direct radiative effect (DRE) of aerosols, which refers to the radiation changes caused by aerosol absorption and scattering, is the primary driver of aerosol–climate interactions. In the weather scale, the aerosol-radiation interaction (ARI) is quite important for predicting air quality, as it can significantly alter the meteorology, affecting visibility and $PM_{2.5}$ concentrations (Peng et al., 2022; Wang et al., 2022a). Including ARI in mesoscale weather forecasts is essential for more accurate radiation flux estimation, particularly during dust storms or haze episodes (Wang et al., 2010, 2015a, b).

Aerosol optical properties, namely, the extinction efficiency ($q_{ext}$), single scattering albedo ($\omega$), and asymmetry factor ($g$), are key parameters for understanding direct aerosol–radiation interactions in weather/climate models. These optical properties of aerosols vary due to multiple factors, including wavelength, complex refractive index, and particle size and morphology. Among these factors, morphological features are particularly challenging to address and can be divided into two aspects: (1) nonsphericity and (2) inhomogeneity. In general, simplified spherical models are commonly used to represent hygroscopic aerosol particles such as sulfate, nitrate, and ammonium, which is consistent with observations in both the field and laboratory (Buseck and Pósfai, 1999; Wise et al., 2007). However, insoluble aerosols like BC and mineral dust aerosols are nonspherical (Adachi et al., 2007; Pósfai et al., 2013). Additionally, BC and mineral dust aerosols are not always homogeneous, as they can mix with other particles during transport, resulting in embedded aerosols within coating materials (Adachi et al., 2010).

Several electromagnetic scattering algorithms are available for calculating the optical properties of aerosol particles with different morphological features. The Lorenz–Mie theory is widely applied for calculating optical properties of homogenous spherical particles (Bohren and Huffman, 1998). The multiple sphere T-matrix method (MSTM) is an implementation of a generalized Mie theory and is suitable for analyzing multiple sphere domains, such as coated spheres and fractal aggregates with spherical monomers (Mackowski, 2014). The discrete dipole approximation (DDA) method is a general scattering algorithm for particles of arbitrary geometry and composition, in which the scatterer is divided into small cubical subvolumes (Yurkin and Hoekstra, 2011). The invariant imbedding T-matrix method (IITM) is a more efficient algorithm for analyzing randomly oriented nonspherical and inhomogeneous particles, which are discretized in terms of multiple inhomogeneous spherical layers (Bi and Yang, 2014; Bi et al., 2013, 2022; Wang et al., 2023c).

The aerosol optical parameterization scheme in climate/weather models is closely tied to the capabilities of electromagnetic scattering algorithms. Spherical aerosol models are commonly used in climate/weather models due to the relatively low computational time required by the Lorenz–Mie theory. Examples of spherical aerosol models include (1) the external-mixing model, where each kind of aerosols is assumed to be a homogeneous sphere, (2) the volume-mixing model,

where all species are homogeneously mixed, (3) the Maxwell-Garnet model, where isolated spherules are suspended in an embedding sphere, and (4) the core-shell model for concentrically coated particles (Bond and Bergstrom, 2006; Fast et al., 2006; Kotchenova and Vermote, 2007). For example, in Weather Research and Forecasting model coupled with Chemistry (WRF-chem), multiple aerosol optical schemes are included, such as the volume-mixing, the Maxwell-Garnet, and the core-shell models (Barnard et al., 2010). In the mesoscale version 5.1 of Global/Regional Assimilation and Prediction System coupled with Chinese Unified Atmospheric Chemistry Environment model (GRAPES_Meso5.1/CUACE), only the external-mixing model is included (Gong and Zhang, 2008; Wang et al., 2018). As a climate model, the Community Atmosphere Model version 5 (CAM5) incorporates both volume-mixing and core-shell treatments in Aerosol Two-dimensional bin module for formation and Aging Simulation (ATRAS) model using look-up tables (Matsui, 2017).

Implementing electromagnetic scattering algorithms in climate and weather models is unrealistic due to the prohibitively long computational time. Therefore, look-up tables with pre-computed optical properties are typically used to incorporate morphologically realistic aerosol models in optical parameterization, reducing computational costs. For example, bare fractal aggregates model and core-grey-shell model have been applied to address the nonsphericity and inhomogeneity of BC using look-up tables in the Multi-scale Atmospheric Transport and Chemistry model (MATCH) and the Sectional Aerosol module for Large Scale Applications (SALSA) (Andersson and Kahnert, 2016). Recently, deep learning has emerged as a new method to handle the optical properties of complex internal mixing models. Wang et al. (2022b) incorporated coated super-spheroid models for mineral aerosols into the WRF-chem model using a fully connected neural network (Wang et al., 2022b).

In this study, we developed a comprehensive AI-based nonspherical aerosol optical scheme (AI-NAOS) to investigate the direct aerosol radiation effects by employing morphologically realistic models based on the deep learning method. In this scheme, insoluble aerosols, such as soil dust (SD) and BC, were both treated as nonspherical cores that are partially or fully embedded in a spherical or nonspherical shell consisting of hygroscopic aerosols. Specifically, we modelled dust as super-spheroids, which have been proved successful in matching the optical properties of dust samples and the polarized radiance observations from Polarisation and Anisotropy of Reflectances for Atmospheric Science coupled with Observations from a Lidar (PARASOL) (Lin et al., 2018, 2021). BC was assumed to exist as fractal aggregates that could accurately reproduce the measured linear backscattering depolarization ratio (Luo et al., 2019; Wang et al., 2023b). We developed optical property databases of these two models using the IITM method. Subsequently, several deep neural networks (DNN) were trained based on the aforementioned databases and then integrated into the AI-NAOS module. The AI-NAOS scheme was online-coupled with the chemical weather models, including WRF-chem and GRAPES_Meso5.1/CUACE.

In the remaining sections of this paper, we will present the details of the AI-NAOS module in Section 2. We will analyze real-case evaluation performed by the GRAPES_Meso5.1/CUACE model in Section 3. Finally, in Section 4, we will summarize and discuss the conclusions.

## 2 Methods

### 2.1 Aerosol optical modelling

In the aerosol module of CUACE model, there are seven aerosol species, including two insoluble aerosols, black carbon and soil dust, and five hygroscopic aerosols, such as organic carbon, sulfate, sea salt, nitrate, and ammonium. The optical properties of these aerosols are accessed through a series of look-up tables, where complex refractive indices are implicitly represented. In the CUACE model, the refractive indices of SD were obtained from the Aeolian dust experiment on climate (ADEC), while the high-resolution transmission molecular absorption database (HITRAN) were applied to represent refractive indices of other aerosols (Rothman et al., 2005; Wang et al., 2006). In the aerosol optical modelling process, aerosol particles of each species were simplified as spheres and completely dissolved in aerosol water, which undergo hygroscopic growth, and then mix externally. This aerosol scheme is a typical external-mixing scheme in which aerosol particles are always assumed to be spherical and homogeneous.

As an update, the AI-NAOS module incorporates improvements related to nonsphericity and inhomogeneity into the aerosol scheme. Additionally, two widely used internal-mixing schemes, the volume-mixing and core-shell methods, have also been integrated into the aerosol module to facilitate a better understanding of the differences between aerosol schemes with various optical models.

Two nonspherical models, fractal aggregates and super-spheroids, were incorporated to represent insoluble aerosols, while hygroscopic aerosols were treated as a coating shell to introduce inhomogeneity. Specifically, the fractal aggregate model composed of identical spheres was found to be a good fit for the morphology of BC particles. The fractal law was used to characterize aggregates, shown below (Sorensen, 2001; Teng et al., 2019):

$$N_s = k_0 \left(\frac{R_g}{R}\right)^{D_f},$$ (1)

In Equation 1, $D_f$ is the fractal dimension, $k_0$ is the scaling prefactor, $N_s$ is the number of spherical monomer, R is the radius of monomer, and $R_g$ is the radius of gyration (Sorensen, 2011). The fractal dimension and the scaling prefactor are major factors influencing the compactness of fractal structure, which is connected with the aging progress of BC (Forrest and Witten, 1979; Li et al., 2016). Tuning a smaller scaling prefactor or fractal dimension allows for the creation of a looser and more linear structure, which shows better fitness for newly emitted BC. Conversely, a larger scaling prefactor and fractal dimension is more suitable for aged BC after transportation. Generally, the fractal dimension ranges from 1.8 to 2.8 (Wang et al., 2017; Wu et al., 2023). In this study, the fractal dimension was set at 2.1, and the scaling prefactor was set at 1.2. A tuneable algorithm called FracVAL was used to generate this fractal aggregates (Morán et al., 2019). The refractive index of BC was assumed to be 1.95 + 0.79i (Bond and Bergstrom, 2006); however, this value can be flexibly adjusted in the module. All hygroscopic aerosols were treated as a homogenous spherical coating shell using the volume-mixing method. The center

of the spherical coating shell was located at the midpoint of the major axis of the fractal aggregates, which was defined based on two spherical monomers with the greatest separations, partially embedding the fractal framework as shown in Figure 1a.

A simplified equation of super-ellipsoid (referred to as super-spheroid) was used for geometric modelling (Barr, 1981; Bi et al., 2018a):

$$\left(\frac{x}{a}\right)^{\frac{2}{n}} + \left(\frac{y}{a}\right)^{\frac{2}{n}} + \left(\frac{z}{c}\right)^{\frac{2}{n}} = 1 \,, \tag{2}$$

in which $n$ is the roundness parameter and $a/c$ is the aspect ratio. In this case, the roundness parameter was fixed at 2.6, which was suitable for optical modelling of SD (Lin et al., 2018). The coating shell was also modelled as a super-spheroid with the roundness parameter ranging from 2.6 to 1.0, for which the core was fully embedded as shown in Figure 1b.

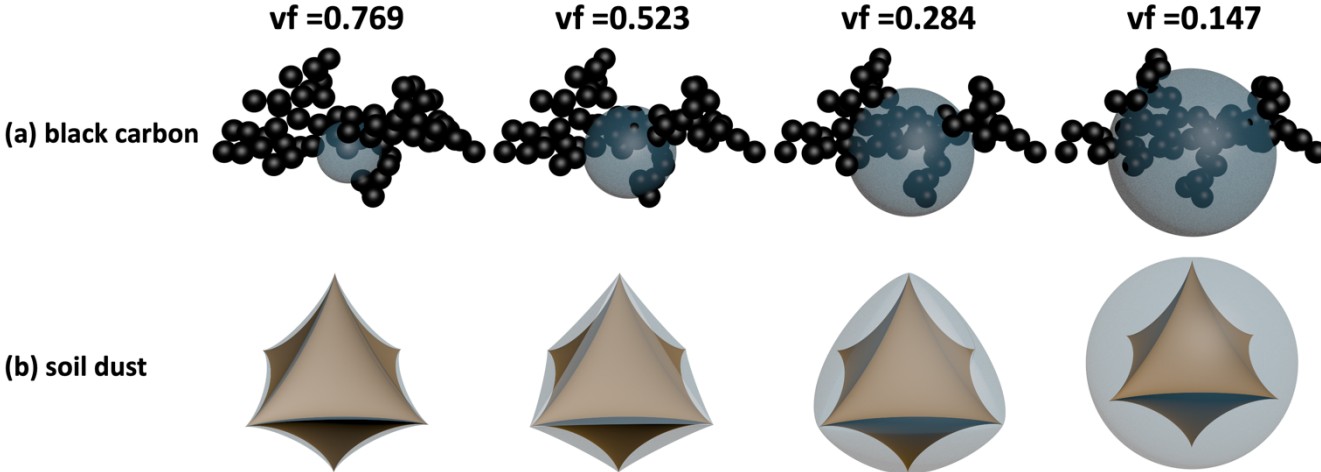

Figure 1. Optical modelling for (a) fractal aggregates framework of black carbon (BC) partially encapsulated with spherical coating
of hygroscopic aerosols and (b) super-spheroid framework of soil dust (SD) fully coated with another super-spheroid of hygroscopic aerosols with various volume fractions.

## 2.2 Database of optical properties

Next, we discuss optical property databases that have been constructed for nonspherical and inhomogeneous particles, specifically encapsulated BC fractal aggregates and coated SD super-spheroids. These databases utilized six parameters to
describe the microphysics of a certain particle, including size parameter, volume fraction (vf), and complex refractive indices of both insoluble aerosols and hygroscopic aerosols. Three optical properties, namely, extinction efficiency ($q_{ext}$), single scattering albedo ($\omega$), and asymmetry factor (g) were determined using these parameters. The volume fraction is a key parameter that describes the mixing status and is defined as shown below:

$$vf = \frac{V_{insol}}{V_{hygro} + V_{insol}} \,, \tag{3}$$

in which $V_{insol}$ is the volume of insoluble aerosols, and $V_{hygro}$ is the volume of hygroscopic aerosols in a single particle. The size parameter (x) is a size defined relative to wavelength and is given by:

$$x = \frac{\pi D}{\lambda},$$ (4)

in which $\lambda$ is the wavelength, and D is the diameter. For nonspherical particles, the actual diameter is the length of the major axis, which can be determined by two monomers with the longest distance in fractal aggregates and the longest dimension in
super-spheroids. Accounting for the convenience of matching with spherical models, the diameter defined in Equation 4 was calculated based on the equi-volume sphere. As shown in Table 1, the size parameter exceeds 20, which is sufficient to handle particles with diameters less than 4 μm even in photosynthetically active radiative bands. The complex refractive indices for hygroscopic aerosols and two insoluble aerosols, BC and SD, are denoted by $mr_{hygro}$ and $mr_{insol}$ for the real prat and $mi_{hygro}$ and $mi_{insol}$ for the imaginary part. The volume fraction of insoluble aerosols ranges from 0.04 to 1.0 for
BC aerosols and from 0.15 to 1.0 for SD aerosols. These parameters are summarized in Table 1.

Table 1. Optical property database of nonspherical and inhomogeneous (NSIH) particles. Black carbon (BC) was modelled as encapsulated fractal aggregates while soil dust (SD) was assumed to be a coated super-spheroid.

| | coated fractal aggregates | coated super-spheroids |
|---|---|---|
| $mr_{hygro}$ | 1.20, 1.25, 1.30, 1.35, 1.40, 1.45, 1.50, 1.55, 1.60 | 1.2, 1.3, 1.4, 1.5, 1.6, 1.8 |
| $mi_{hygro}$ | 0, 0.001, 0.005, 0.01, 0.05, 0.1 | $10^{-5}$, $10^{-4}$, 0.001, 0.005, 0.01, 0.05, 0.1, 0.5 |
| vf(%) | 3.66, 5.42, 8.46, 11.79, 16.77, 24.3, 35.82, 43.28, 52.03, 61.74, 71.57, 80.78, 88.89, 94.99, 100 | 14.68, 18.06, 22.51, 28.35, 36.02, 46.11, 59.38, 76.88, 100 |
| $mr_{insol}$ | 1.65, 1.75, 1.85, 1.95 | 1.2, 1.4, 1.5, 1.6 |
| $mi_{insol}$ | 0.5, 0.6, 0.7, 0.8 | 0.0005, 0.001, 0.005, 0.01, 0.05, 0.1, 0.5 |
| x | 0.1-21.0 | 0.05-26.4 |

## 2.3 Bulk optical properties

  In the aerosol module of the CUACE, aerosol particles are categorized into 12 size bins based on their diameter,
ranging from 0.01 μm to 40.96 μm. When utilizing this model, it is necessary to integrate the optical properties over the diameter within each size bin to obtain bulk optical properties that are more representative, as the oscillation of optical properties along with the particle size occurs. To achieve this, we used the log-normal size distribution for each size bin, ranging from the lower bound $D_{min}$ to the upper bound $D_{max}$. The probability density function (PDF) of log-normal distribution is defined as shown below:

$$n(D) = \frac{1}{D\sqrt{2\pi\sigma}} e^{-\frac{(lnD - lnD_m)^2}{2\sigma^2}},$$ (5)

in which $\sigma$ is the standard deviation, and $D_m$ is the mean diameter. Based on the PDF, bulk optical properties can be calculated by integration as shown below:

$$Q_{ext} = \frac{\int_{D_{min}}^{D_{max}} q_{ext}\, D^2 n(D) dD}{\int_{D_{min}}^{D_{max}} D^2 n(D) dD}, \tag{6}$$

$$<SSA> = \frac{\int_{D_{min}}^{D_{max}} \omega q_{ext}\, D^2 n(D) dD}{\int_{D_{min}}^{D_{max}} q_{ext} D^2 n(D) dD}, \tag{7}$$

$$<G> = \frac{\int_{D_{min}}^{D_{max}} g \omega q_{ext}\, D^2 n(D) dD}{\int_{D_{min}}^{D_{max}} \omega q_{ext} D^2 n(D) dD}, \tag{8}$$

in which $Q_{ext}$, <SSA> and <G> are the bulk extinction efficiency, bulk single scattering albedo and bulk asymmetry factor, respectively.

**2.4 Deep Neural Networks**

Aerosol optical parameterizing is a complex task that involves various species and multiple processes, such as aerosol mixture and hygroscopic growth, that need to be addressed. In the past, the look-up table method was widely used for this purpose, but it had limitations in terms of simplifying some processes and consuming a relatively large amount of storage space. To overcome these challenges, a deep learning method can be applied. A deep neural network (DNN) can efficiently and accurately provide continuous predictions while requiring less storage space compared to a look-up table (Yu et al., 2022a, b). Additionally, the DNN method eliminates the need for multiple interpolations among various dimensions, making it easier to compute the optical properties at the particle size, volume fraction, and complex refraction indices based on the real-time status of aerosols. A multiple-target DNN model that has been designed to infer the single-scattering properties of encapsulated fractal aggregates of BC was adapted to bulk optical properties inference in this study. (Wang et al., 2023b). Four DNNs have been trained based on bulk optical property databases of spherical and NSIH particles using this architecture, including spheres, core-shell spheres, encapsulated fractal aggregates and coated super-spheroids. Note that, the present DNNs are different from those in previous studies, in which DNNs are obtained in terms of the single-scattering properties for general application purposes. More details regarding the multiple-target DNNs are provided in the Appendix.

**2.5 Nonspherical Aerosol Optical Scheme**

A basic process in the AI-NAOS module is the determination of particle states, including the volume of each kind of particle, volume fraction of mixed particles, and the morphology of particles. Coupled with a size-segregated aerosol module, the particle states are independent among every size bin. In a specific bin, all hygroscopic aerosols are first internally mixed with aerosol water based on a volume-mixing assumption and then encapsulated other two insoluble

aerosol, respectively. The key issue is to determine the amount of hygroscopic aerosols coating, i.e., the volume fraction. The simplest method is to make all hygroscopic aerosols be the coating, like the core-shell scheme, which means the volume fraction ranges from zero to one. A more appropriate method is to consider a lower limit of the volume fraction, based on worldwide field observations (Wang et al., 2023a). Specifically, the lower limit of the volume fraction was set to be 0.3 for BC and 0.6 for SD. Thus, not all hygroscopic aerosols were internally mixed with insoluble aerosols when the volume of hygroscopic aerosols was large enough. These extra mixed hygroscopic aerosols were assumed to be homogenous spheres. Due to hygroscopic growth, the diameters of different hygroscopic aerosol species were also different. Similar to the core-shell method, the diameter of internally mixed particle was determined using the volume-weighted method. In this way, all seven aerosol species could be treated as three kinds of internally-mixed particles, including two nonspherical and inhomogeneous particles (partially encapsulated fractal aggregates of BC, and fully coated super-spheroid of SD) and the homogeneous and spherical particle of hygroscopic aerosols.

Once the state of aerosols was determined, the optical properties of three internally-mixed aerosol particles were inferred by the DNNs integrated in the AI-NAOS module. The total optical properties were calculated based on the external-mixing assumption, in which extinction and scattering coefficients were the sum of each species. This implementation of the AI-NAOS module provided an efficient estimation of optical properties for nonspherical and inhomogeneous particles. Generally, three optical properties (extinction efficiency, single-scattering albedo, and asymmetry factor) were considered sufficient for radiation transfer calculation in the chemical weather model. The framework of AI-NAOS module was illustrated in Figure 2.

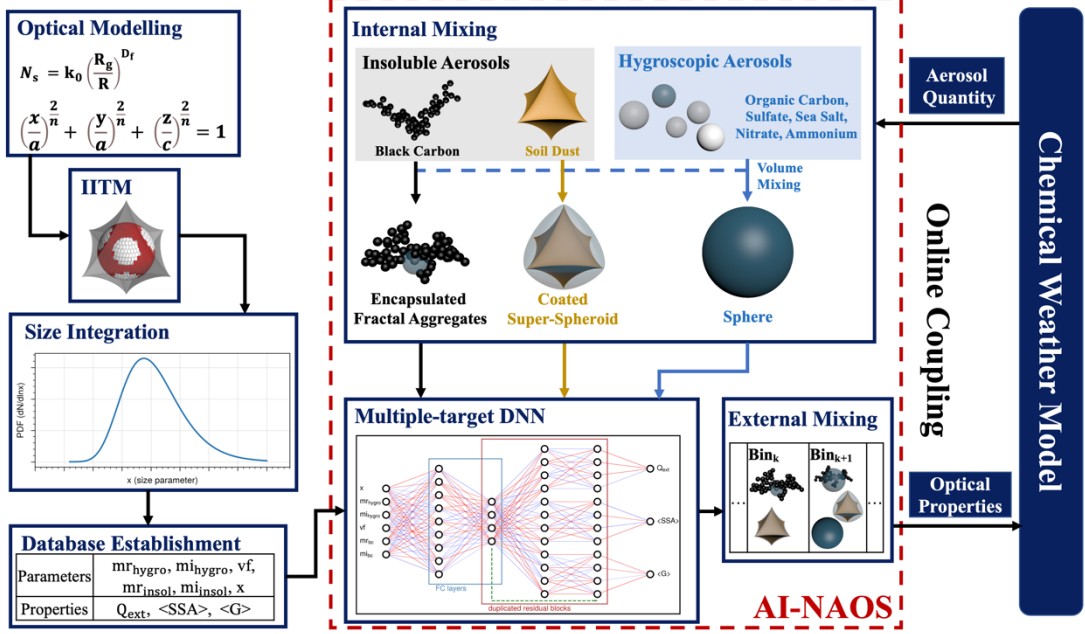

**Figure 2. The framework of AI-NAOS module.**

The bulk optical properties were compared and shown in Figure 3 for different aerosol optical schemes, including the external-mixing, volume-mixing, core-shell, and AI-NAOS modules. The AI-NAOS module contained encapsulated fractal aggregates model of BC and coated super-spheroids of SD. The calculations of bulk optical properties were performed considering the presence of hygroscopic aerosols and either BC or SD as the insoluble aerosol species. The volume fraction of insoluble aerosols was assumed to be 0.33. The complex refractive indices of BC, SD, and hygroscopic aerosols were 1.95+0.79i, 1.5+0.01i, and 1.4+0.0001i, respectively. Generally, BC aerosols are small particles with diameters around 0.2 μm, and size parameters around 0.9–1.6 for photosynthetically active radiative bands (Wu et al., 2023). As shown in the top row of Figure 3, the extinction efficiency of the AI-NAOS scheme was smaller compared to the core-shell scheme and the volume-mixing scheme, but larger compared to the external-mixing scheme. As for SSA, it could be observed that the external-mixing scheme yielded the largest results, while the other three schemes were close. It is likely that the AI-NAOS scheme led to improvements in the estimation of BC absorption, as the external-mixing scheme underestimated BC absorption and the volume-mixing scheme overestimated this parameter (Kahnert et al., 2012). The asymmetry factor calculated by the AI-NAOS was slightly higher than other schemes, leading to a smaller backscattering fraction. With respect to SD aerosol, the aerosols' diameters were much larger, so we focus on the part where the size parameter was larger than 10.0, as shown in the bottom row in Figure 3. The extinction efficiency based on the AI-NAOS scheme was larger, and the SSA was similar compared to other schemes for large SD particles, indicating stronger absorption due to the NSIH effect of the SD. The asymmetry factor of the AI-NAOS scheme was the smallest.

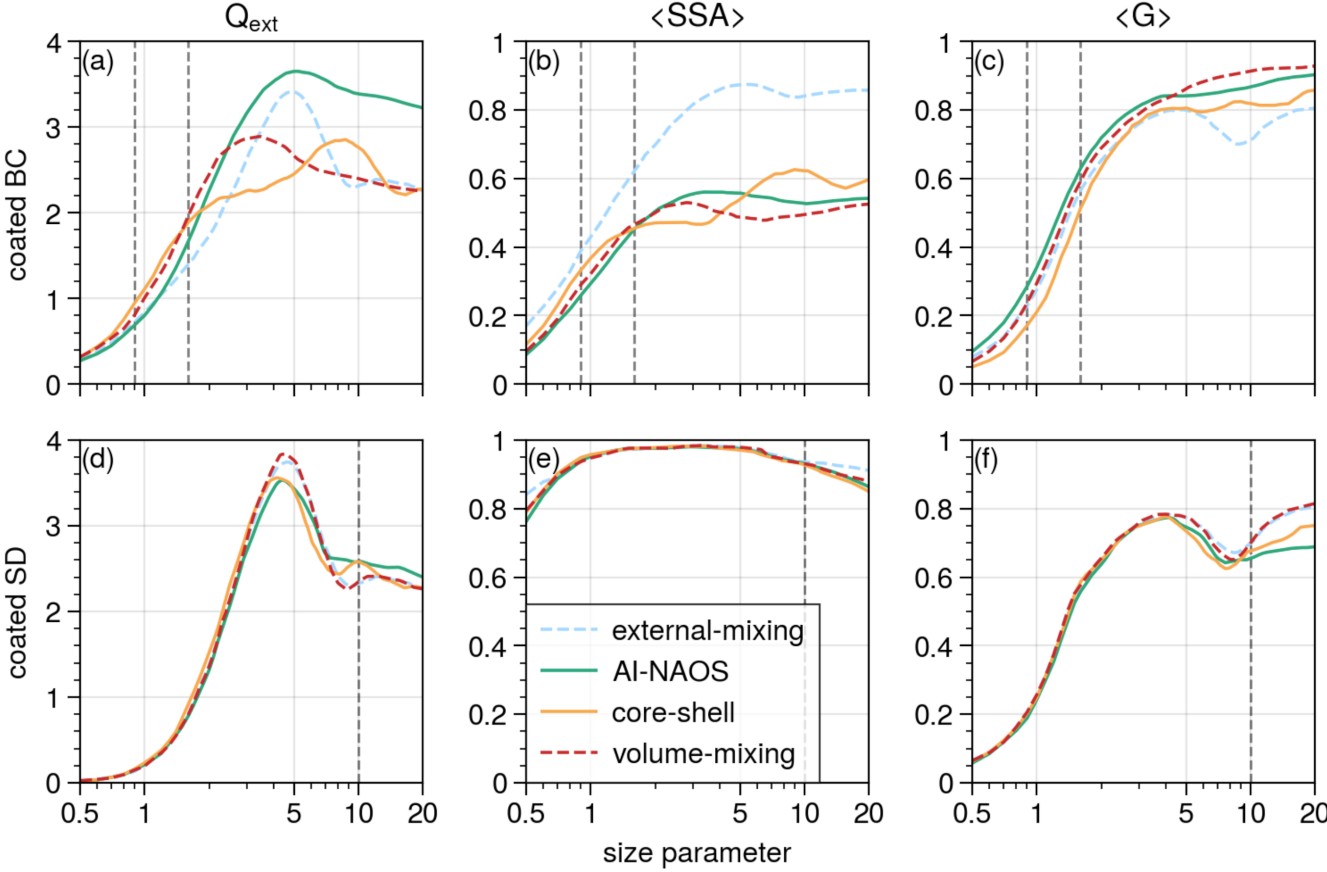

**Figure 3. Optical properties of (a-c) black carbon (BC) and (d-f) soil dust (SD) with four optical modelling schemes (external-mixing, AI-NAOS, core-shell, and volume-mixing scheme).**

## 2.6 Model configuration

The AI-NAOS module was on-line coupled with chemical weather model GRAPES_Meso5.1/CUACE. In a previous study, the NSIH effect of SD had been evaluated using the WRF-chem model, revealing that the nonsphericity of SD could cause more surface dimming and more solar heating (Wang et al., 2022b). At that time, BC was still assumed to be spherical and treated using the core-shell model. Thus, this study primarily focuses on the NSIH effects of BC, where various models of BC were compared while SD was always assumed to be super-spheroids. Real-case simulations were conducted using GRAPES_Meso5.1/CUACE to gain a better understanding of the impact of aerosols on the weather. Five different aerosol treatments were applied in the simulations, including a control experiment without aerosol optical effects, the AI-NAOS scheme, the core-shell scheme with spherical but inhomogeneous model, and the external-mixing and the volume-mixing schemes for the spherical and homogeneous model. To assess the NSIH effect from the BC aerosol, different models were applied to only the BC aerosols, while the SD aerosols were modelled as super-spheroids among all schemes to avoid the

influence due to SD. The AI-NAOS scheme was directly connected with the shortwave radiation scheme (Goddard Space Flight Center Scheme);namely, the primary driver of NSIH effect on aerosol–weather interactions was a perturbance in the solar radiation.

Real-scene studies were conducted to investigate the NSIH effect on the thermodynamic structure of the atmosphere under a heavy pollution scenario. The case was carried out on January 12, 2018, running for 72 hours, with the last 48 hours used for evaluation. The simulation domain covered East China (95–125°E, 20–50°N) with a horizontal resolution of 0.1° × 0.1°. The evaluations mainly focused on three regions with high emissions: (1) Sichuan Basin (103.5-108.5°E, 28.5-32.5°N), (2) Middle Yangtze Plain (110.5–114.5°E, 27.5–32.5°N), and (3) North China Plain (114-120°E, 33-39°N). The initial meteorological field and lateral boundary conditions were constructed using the NCEP FNL data on 0.25 × 0.25 grids (National Centers For Environmental Prediction/National Weather Service/NOAA/U.S. Department Of Commerce, 2015). The Multi-resolution Emission Inventory model for Climate and air pollution research (MEIC) was used for chemical emissions (Li et al., 2017).

The configurations of physical and chemical processes in GRAPES_Meso5.1/CUACE simulations are summarized in Table 2.

**Table 2. Physical and chemical processes in the GRAPES_Meso5.1/CUACE.**

| Process | Scheme |
| --- | --- |
| Microphysics | Thompson (Thompson et al., 2004) |
| Shortwave radiation | Goddard (Chou et al., 1998) |
| Longwave radiation | RRTM (Mlawer et al., 1997) |
| Surface layer | SFCLAY (Pleim, 2007) |
| Land surface | Noah (Chen et al., 1997) |
| Planetary boundary layer | MRF (Hong and Pan, 1996) |
| Cumulus | KFETA (Kain and Fritsch, 1993) |
| Gas-phase chemistry | RADM2 (Stockwell et al., 1990) |
| Aerosol | CUACE (Zhou et al., 2012) |
| Aerosol optics | 1. NA (no aerosol optical effect)<br>2. external-mixing<br>3. AI-NAOS<br>4. core-shell<br>5. volume -mixing |

## 3 Result

### 3.1 Performance of Deep Neural Networks

Figure 4 illustrates the performance of DNNs for these two NSIH models, and three statistical metrics, namely, mean absolute error (MAE), root mean square error (RMSE), and coefficient of determination (R-squared), were used to evaluate the DNN. Approximately 100,000 samples were randomly selected from the entire database (training, validation, and testing sets) for each optical property of each NSIH model. The scatter points in Figure 4 represent the evaluation results of the DNN predictions and IITM calculations, with the color denoting the probability density scaled from 0 to 1. For coated fractal aggregates of BC, the RMSE of bulk extinction efficiency, bulk SSA, and bulk asymmetry factor were 0.0114, 0.0017, and 0.0013, respectively. For coated super-spheroids of SD, the RMSE values were 0.0240, 0.0062, and 0.0052, respectively. The R-squared values for all three optical properties were above 0.99, indicating a good fit between the true values from the IITM calculations and the predicted values from the DNNs. Overall, the optical properties of the NSIH models can be accurately predicted using the DNN models.

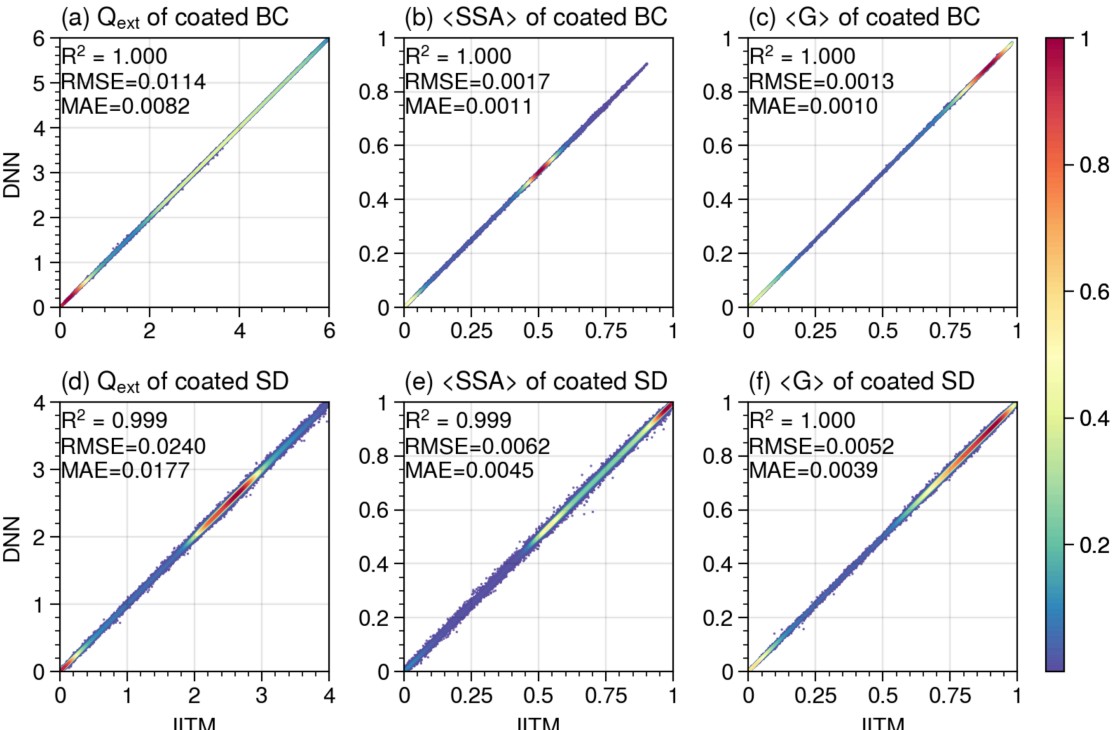

**Figure 4. Comparison of optical properties calculated using the invariant imbedding T-matrix method (IITM) and predictions from deep neural networks (DNNs). (a–c) encapsulated fractal aggregate model for black carbon. (d–f) coated super-spheroid model for soil dust. The probability density is scaled from 0 to 1.**

## 3.2 Aerosol optical properties

It is essential to evaluate three bulk aerosol optical properties ($Q_{ext}$, <SSA>, and <G>), which are refined directly in the AI-NAOS module. To effectively display these properties of the vertical atmospheric column, the column aerosol optical properties, which include aerosol optical depth (AOD), absorbing aerosol optical depth (AAOD), and column asymmetry factor (G), are defined as follows(Taylor, 2012):

$$b_{ext} = \sum_{i=1}^{12} \sum_{j=1}^{3} Q_{ext_{ij}} \frac{\pi}{4} D_{ij}^2 N_{ij} ,\tag{9}$$

$$AOD = \int_{z=0}^{H_{TOA}} b_{ext} dz ,\tag{10}$$

$$AAOD = \int_{z=0}^{H_{TOA}} b_{ext}(1-<SSA>)dz ,\tag{11}$$

$$G = \frac{\int_{z=0}^{H_{TOA}} b_{ext}<SSA><G>dz}{\int_{z=0}^{H_{TOA}} b_{ext}<SSA>dz} ,\tag{12}$$

where i is the index of size bin, j is the index of aerosol particle, N is the number concentration of aerosol particles, $b_{ext}$ is the extinction efficiency, z is the height above ground, and $H_{TOA}$ is the height of TOA.

Figure 5 illustrates three column optical properties at 0.55 μm based on the AI-NAOS scheme, as well as anomalies based on the other three aerosol optical schemes. All properties were averaged over 48 hours from the 25[th] to the 72[th] hour to yield a general picture. Three distinct centers with extreme AOD values, Sichuan Basin, Middle Yangtze Plain, and North China Plain, are marked with black rectangles from west to east. These three regions also stood out as major sources of aerosol emissions. The North China Plain was characterized by intense industrial manufacturing while residential activities

and transportation played prominent roles in the Sichuan Basin and Middle Yangtze Plain compared to the surrounding areas. Within the AI-NAOS module, the spatial average of AOD was 0.29, 0.37, and 0.36 over the Sichuan Basin, Middle Yangtze Plain, and North China Plain, respectively. The AOD values were larger compared to the external-mixing scheme but slightly smaller compared to the core-shell and volume mixing schemes. With respect to AAOD, the values were found to be 0.06, 0.08, and 0.08 averaged over the three marked regions, from west to east, respectively. Taking the results under

the AI-NAOS module as a benchmark, the AAOD values based on three spherical schemes (the external-mixing, core-shell, and volume-mixing scheme) were 70.4%, 125.3%, and 129.3% over Sichuan Basin, 70.0%, 119.6%, and 126.1% over the Middle Yangtze Plain and 70.4%, 116.0%, and 124.4% over the North China Plain, respectively. The column asymmetry factor is illustrated in the bottom panel of Figure 5. The values based on the AI-NAOS module was 0.66, 0.64, and 0.63 averaged over the three marked regions, which was roughly +0.03, +0.03, and +0.01 larger compared to the three spherical

schemes.

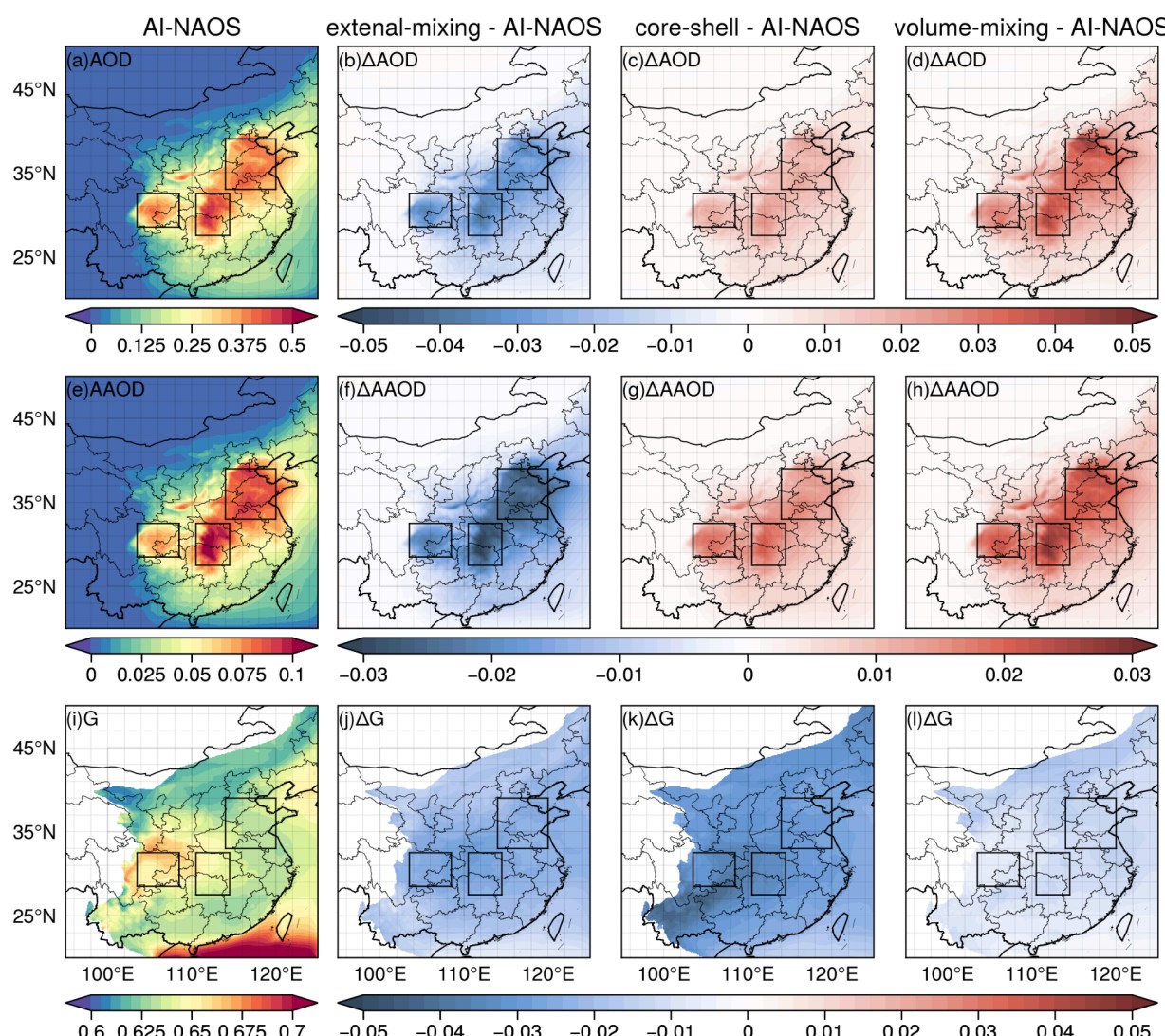

**Figure 5. Aerosol column optical properties based on the AI-NAOS scheme and anomalies based on three spherical schemes (external-mixing, core-shell, and volume-mixing).**

The spatially averaged column optical properties of the three spherical schemes, all of which were benchmarked to values under the AI-NAOS module, are summarized in Table 3. It is clear that there was minimal variation in the asymmetry factor, whereas a significant disparity existed in the AAOD. The AAOD, indicating the aerosol absorption, is greatly affected by aerosol optical modelling. Specifically, the external-mixing scheme, lacking both nonspherical and inhomogeneous effects, estimated notably weaker absorption, while the core-shell model, only considering inhomogeneous effects, estimated stronger absorption. The AI-NAOS module is anticipated to correctly deal with the aerosol absorption, yielding a moderate estimation.

**Table 3. Aerosol optical properties of three spherical schemes averaged over three marked regions, benchmarked to values under the AI-NAOS module.**

| Optical Properties | Regions | External-mixing | Core-shell | Volume-mixing |
|---|---|---|---|---|
| AOD | Sichuan Basin | 91.7% | 104.9% | 107.0% |
| | Middle Yangtze Plain | 91.7% | 105.1% | 108.0% |
| | North China Plain | 92.3% | 104.8% | 108.4% |
| AAOD | Sichuan Basin | 70.4% | 125.3% | 129.3% |
| | Middle Yangtze Plain | 70.0% | 119.6% | 126.1% |
| | North China Plain | 71.4% | 116.0% | 124.4% |
| G | Sichuan Basin | 96.2% | 94.9% | 98.8% |
| | Middle Yangtze Plain | 96.0% | 94.8% | 98.3% |
| | North China Plain | 96.2% | 95.4% | 98.0% |

As shown in Figure 6, we have used the daily AOD product from the Moderate Resolution Imaging Spectroradiometer (MODIS) to validate our simulations on January 13. The spatial distribution of AOD observed by MODIS exhibits a similar pattern to our simulations, with high AOD values detected over three regions characterized by high anthropogenic emissions: the Sichuan Basin, the Middle Yangtze Plain, and the North China Plain.

To gain a better understanding of AOD distribution pattern, we calculated the probability distribution function (PDF) over a wide region where high AOD values were observed (105–118°E, 27–40°N). Due to the presence of missing values in the MODIS AOD product, corresponding values in our simulations were also omitted. The simulations reveal a more concentrated distribution pattern, with the highest AOD values being slightly lower than those observed by MODIS. The median values of simulated AOD within external-mixing, AI-NAOS, core-shell, and volume-mixing schemes are 0.206, 0.225, 0.232, and 0.238, respectively. The MODIS median value of 0.217 falls between the external-mixing and AI-NAOS schemes, indicating a good agreement between our simulations and the observations.

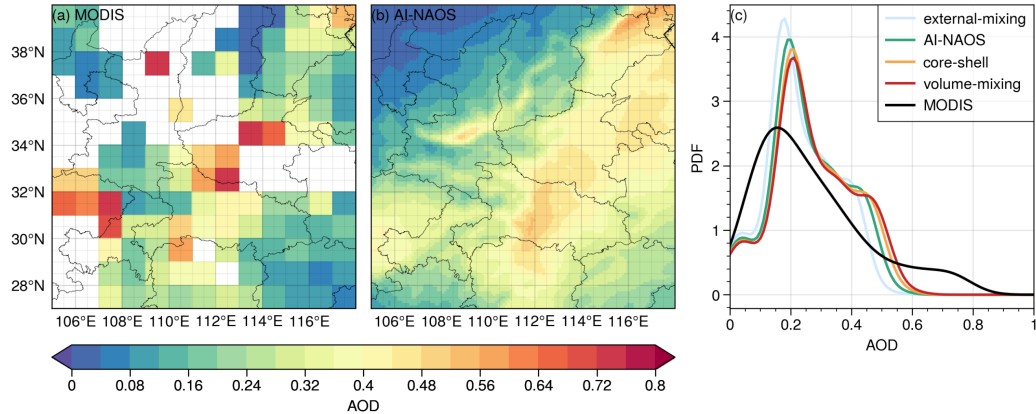

**Figure 6. Spatial distribution of daily AOD on January 13 from (a) MODIS product [MOD08_D3] and (b) simulations with AI-NAOS. (c) AOD probability distribution function of the AOD product and simulations with external-mixing, AI-NAOS, core-shell, and volume-mixing schemes.**

The absorption enhancement (E$_{abs}$), defined as the ratio of absorption section of coated and bare BC particles, is an important metric of aerosol absorption. A significant issue identified is that field observations typically reveal smaller E$_{abs}$ values than those predicted by model simulations. To mitigate this discrepancy, a series of modifications aimed at managing the E$_{abs}$ value were conducted, drawing on microphysical complexities such as coating thickness and morphology (Chen et al., 2023; Huang et al., 2023). However, these strategies were rooted in the core-shell model, utilizing look-up tables or empirical formulas without accounting for aerosol nonsphericity. This limitation potentially leads to inaccuracies in estimations of aerosol absorption.

Figure 7 illustrates the distribution of absorption enhancement and single scattering albedo over the Sichuan Basin. The E$_{abs}$ value was the ratio of AAOD under the external-mixing scheme and other internally-mixed schemes. The median values of E$_{abs}$ under the AI-NAOS, core-shell, and volume-mixing schemes were 1.42, 1.79, and 1.86, respectively. It is clear that the encapsulated fractal models in the AI-NAOS module could induce less absorption enhancement as the lensing effect was weak, as the BC core was only partially coated by non-absorptive hygroscopic aerosols. The median values of single scattering albedo were 0.81, 0.78, and 0.78 under the three schemes, which could also be attributed to the lensing effect. The E$_{abs}$ value could be extremely large (1.80 ± 0.29) when the BC was heavily coated and dropped down to 1.22 ± 0.09 with thin coating (Zhai et al., 2022). In most cases, the mean value of E$_{abs}$ observed in China was quite small, ranging from 1.17 to 1.50 (Ma et al., 2020; Sun et al., 2021; Wu et al., 2018; Zheng et al., 2022). Taking all factors into consideration, the AI-NAOS results, which provided a moderate estimation of AAOD while maintaining a lower E$_{abs}$ value, appeared to be a favorable choice.

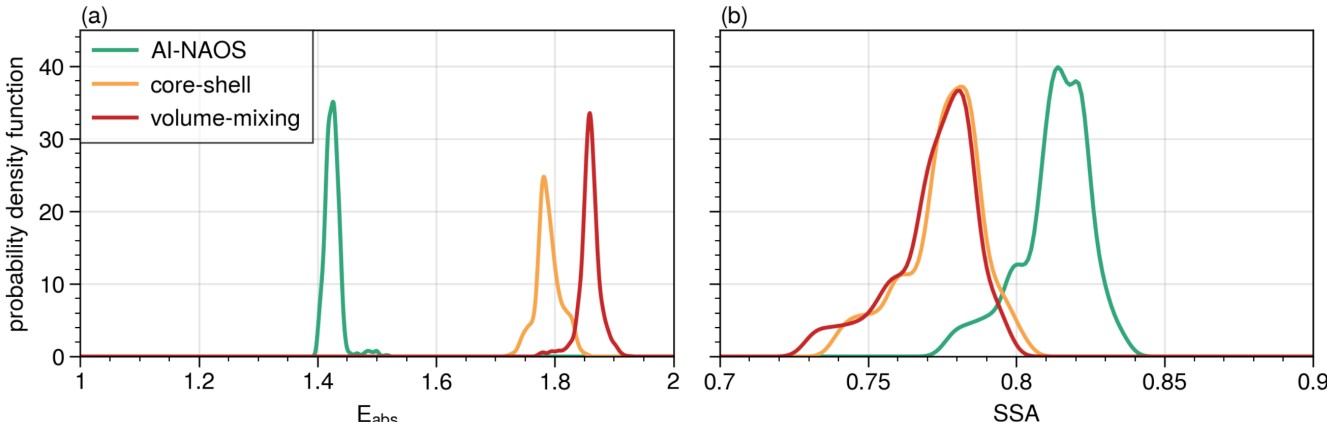

**Figure 7. Probability density function of absorption enhancement and single scattering albedo over the Sichuan Basin within the AI-NAOS, core-shell, and volume-mixing schemes.**

### 3.3 Shortwave direct radiation effect

The shortwave radiative flux is directly influenced by the optical properties of the atmospheric column. Figure 8 depicts the aerosol shortwave DRE at the TOA, within the atmosphere, and at the surface, based on four aerosol optical schemes

(external-mixing, AI-NAOS, core-shell and volume-mixing). Similar to Section 3.1, these DREs were averaged over a 48-hour period.

At the TOA, the spatially averaged DREs based on the AI-NAOS module were +1.9, -2.8, and -1.2 W/m$^2$ over the three
marked regions, respectively. The external-mixing scheme yielded values of +0.3, -4.6, and -3.0 W/m$^2$, while the core-shell scheme showed values of +2.4, -2.8, and -1.2 W/m$^2$, and the volume-mixing scheme showed values of +3.4, -1.8, and -0.4 W/m$^2$. It is observed that aerosols exhibited a warming impact on the Earth's climate system over the Sichuan Basin, whereas a cooling effect was discovered over the Middle Yangtze Plain and North China Plain. In comparison to the AI-NAOS scheme, the warming effect was much weaker with the external-mixing scheme but more pronounced with the core-
shell and volume-mixing schemes.

At the surface, the spatially averaged DREs based on the AI-NAOS module were -21.5, -37.8, and -31.2 W/m$^2$ over the three regions, respectively, indicating a cooling effect of aerosols at the surface. This cooling effect was weaker with the external-mixing scheme and stronger with the core-shell scheme and volume-mixing schemes. This finding can be attributed to atmospheric absorption, which was determined by the difference between net solar fluxes of TOA and the surface. It is
evident that the changes in atmosphere absorption based on the AI-NAOS scheme were extremely large over the three regions, with values of +23.4, +35.0, and +30.0 W/m$^2$, respectively. As shown in the second panel of Figure 8, the atmosphere absorption based on the three spherical schemes were also benchmarked to the AI-NAOS scheme, with values of 80.7%, 112.5%, and 118.9% over Sichuan Basin, 79.6%, 109.2%, and 115.7% over Middle Yangtze Plain, and 80.2%, 105.8%, and 114.3% over North China Plain.

Taking into consideration of the NSIH effect, a warming effect on the Earth's climate system and a cooling effect on the surface were found compared to the external-mixing scheme. Specifically, the DRE induced by the NSIH effect was +1.6, +1.8, and +1.8 W/m$^2$ at TOA and -2.9, -5.3, and -4.1 W/m$^2$ at surface, over the three regions, respectively. Furthermore, these effects can be strengthened using the volume-mixing scheme, which may be too strong and lead to an overestimation.

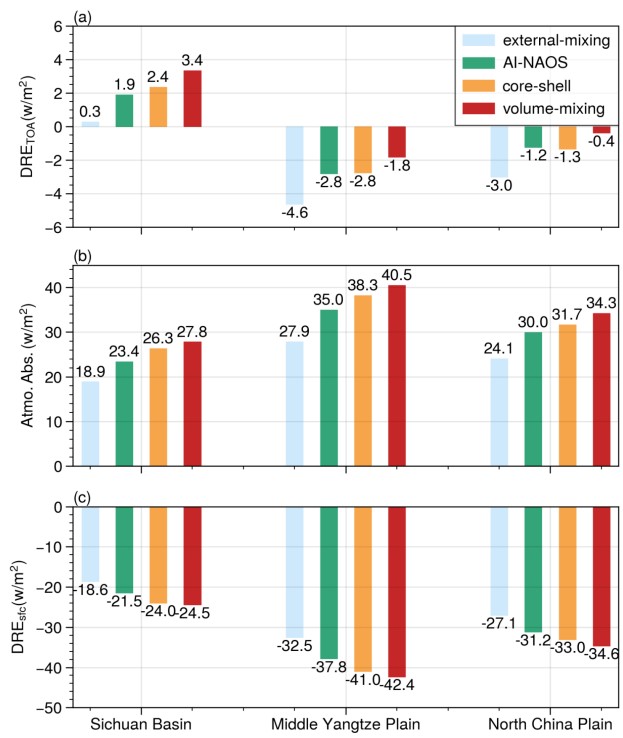

**Figure 8. Aerosol shortwave direct radiation effect based on four aerosol optical schemes (external-mixing, AI-NAOS, core-shell, and volume-mixing) (a) at the top of atmosphere (TOA), (b) within the atmosphere, and (c) at the surface.**

### 3.4 Thermodynamic structure

Solar radiation directly heats the Earth's system and influences the thermodynamic structure of the atmosphere. Scattering aerosols, such as sulfate, can partially offset the warming effects from greenhouse gases, while absorptive
aerosols, such as BC, can lead to strong warming effects (Kaufmann et al., 2011; Ramanathan and Carmichael, 2008). The overall aerosol DRE could result in a spatially averaged surface temperature decrease of -0.487K over China (Persad and Caldeira, 2018), with cooling reaching up to -0.7K in specific regions like the Sichuan Basin (Giorgi, 2002).

Figure 9 illustrates anomalies in temperature at 700 hPa, the height of the planetary boundary layer (HPBL), and the 2-meter temperature. As shown in the first column of Figure 9, the aerosol DRE induced a warming effect at 700 hPa over the
planetary boundary layer (PBL) and a cooling effect at 2 meters above surface, leading to a more stable atmosphere layer and a decrease in $H_{PBL}$. Within the AI-NAOS module, anomalies in temperature at 700 hPa were +0.17, +0.12, and +0.14 K, while anomalies in the 2-meter temperature were -0.83, -1.11, and -1.37K over the three regions, respectively. The values of decreases in $H_{PBL}$ were -75, -108, and -90 meters over the three regions, respectively.

The warming effect at 700 hPa and the cooling effect on the ground were less prominent based on the external-mixing
scheme. The temperature anomalies between the two schemes were +0.04, +0.03, and +0.02 K at 700 hPa and -0.10, -0.14, and -0.17 K on the ground, over the three regions, respectively. The NSIH effect is observed to enhance the temperature

anomalies by at least 21% at 700 hPa and 13% on the ground. Similarly, the $H_{PBL}$ could be further decreased by NSIH effect, with values of -11, -15, and -12 meters over the three regions, respectively. Generally, the anomalies in temperature and $H_{PBL}$ could be more intensified by the core-shell and volume-mixing schemes, which exhibited larger DRE anomalies.

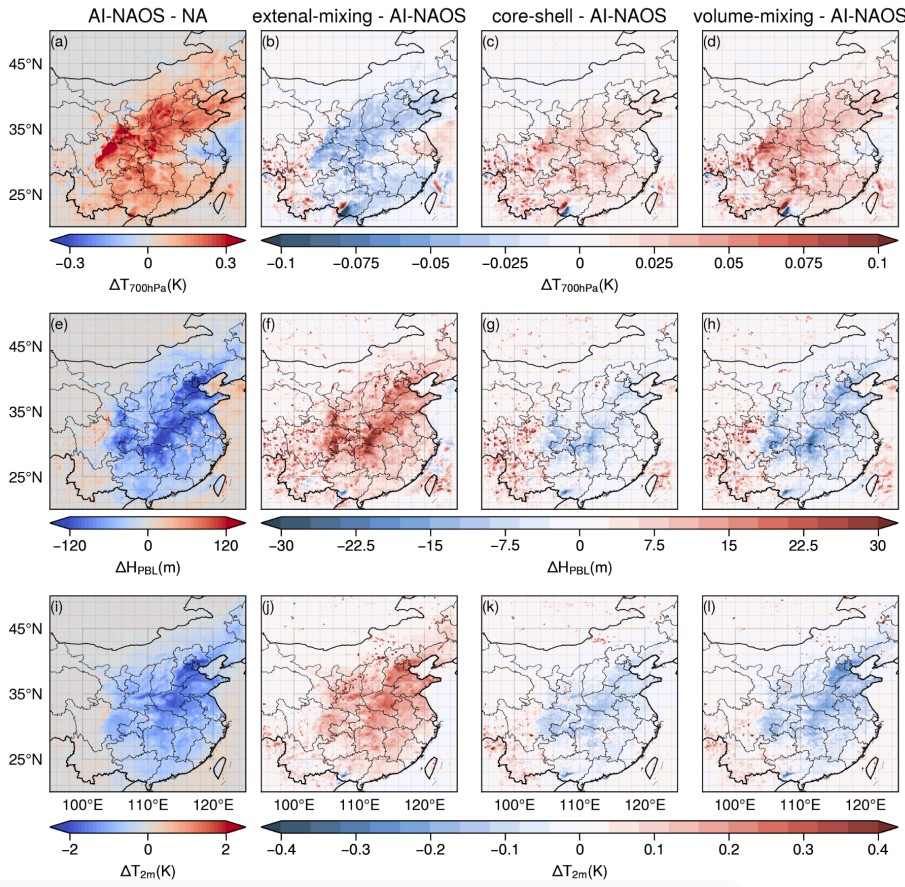

**Figure 9. Anomalies in temperature at 700 hPa, height of planetary boundary layer, and 2-meter temperature caused by the direct radiative effect (DRE). The anomalies are calculated as the difference between the results under the AI-NAOS module and the control scheme without aerosol effects (first column), as well as the difference between the results under three spherical schemes and the AI-NAOS module (second to fourth columns).**

Figure 10 illustrates the vertical profiles of perturbations in temperature and aerosol short-wave radiation heating rate (HR) based on the four optical aerosol schemes, averaged over three specific areas. The aerosol heating effect decreases with the height as aerosol concentrations are higher near the emission source on lower levels. The HR values within the AI-NAOS module were 0.64, 0.72, and 0.66 K/day at the surface, and 0.29, 0.34, and 0.31 K/day at 700 hPa, over the three specific areas, respectively. When vertically averaged, the values within the AI-NAOS module were 0.45, 0.51, and 0.46 K/day, which were +0.09, +0.10, and +0.09 K/day higher than the values within the external-mixing scheme, over the three specific areas, respectively. Therefore, the NSIH effect could enhance the heating rate by at least 23%. This enhancement could be

even stronger within the core-shell and volume-mixing schemes, reaching 30% and 41%, respectively, over the Sichuan Basin.

Temperature perturbations increased with height, indicating a cooling effect at the surface and a warming effect above the PBL. Vertically averaged temperature perturbations under the AI-NAOS module were -0.31, -0.31, and -0.34 K across the three regions, respectively, indicating an overall cooling effect. This cooling effect of the three spherical schemes was 96%, 113%, and 108% compared to the AI-NAOS module over the Sichuan Basin, respectively. In general, NSIH effects induce direct changes in the heating rate based on anomalies of atmosphere absorption, further influencing temperature and thermodynamic structures.

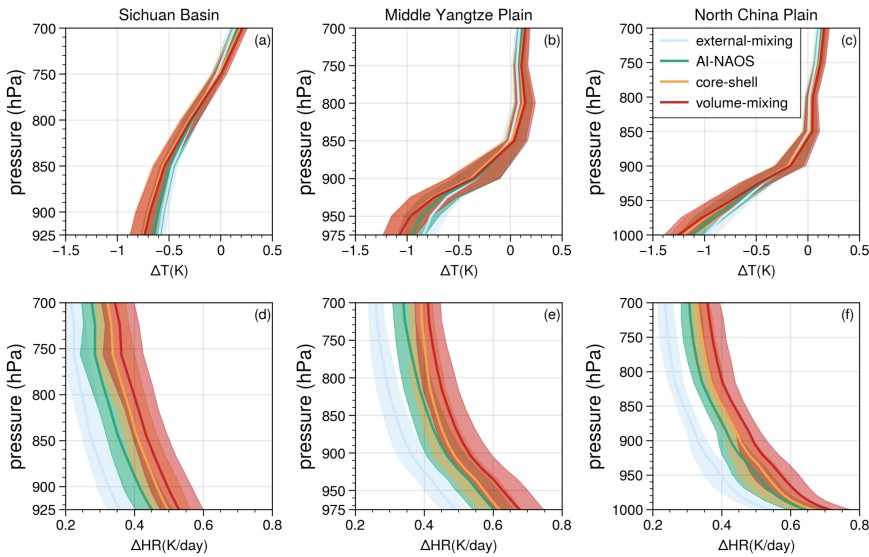

**Figure 10. Vertical profiles of (a-c) temperature and (d-f) short-wave heating rate anomalies based on four aerosol optical schemes (external-mixing, AI-NAOS, core-shell, and volume-mixing). The solid lines represent the median value and the shaded areas encompass the range from the 25 to 75 percentage.**

### 3.5 Precipitation

We then examined the impact of aerosol-induced changes in thermodynamic structure on precipitation anomalies. The direct radiation effect plays a role in reducing solar radiation flux, temperature and moisture fluxes at the surface, leading to the inhibition of convection and a decrease in precipitation (Nabat et al., 2015). Additionally, the semi-direct effect, caused by aerosol absorption heating the lower atmosphere, results in increased atmospheric stability and suppression of convection within the PBL, while inducing stronger convection above this boundary layer (Allen et al., 2019). Collectively, these two effects contribute to reduced precipitation in East Asia. However, anomalous circulations driven by direct radiation effect can impact moisture transport, leading to enhanced precipitation in South China but reduced precipitation in North China (Chen et al., 2018; Huang et al., 2007).

Figure 11 depicts the anomalies in height of PBL and accumulated precipitation over the last 48 hours, comparing the AI-NAOS scheme with a control scheme of zero AOD. The analysis focused on a specific region (101.5-112.5°E, 29.5-34.5°N), encompassing parts of the Sichuan Basin and the Middle Yangtze Plain. The results indicate a suppression effect on precipitation accompanied by a decrease in PBL height in this region. The spatially averaged precipitation anomaly was -0.24 mm for the AI-NAOS module, with anomalies of -0.21, -0.26, and -0.27 mm for the external-mixing, core-shell, and volume-mixing schemes, respectively. The PBL height anomalies for AI-NAOS and the three spherical schemes were -56.8, -48.7, -63.0, and -64.4 meters, respectively. Notably, the suppression effect was less significant in the external-mixing scheme. The NSIH effect amplified the suppression effect by approximately 15%. In the core-shell and volume-mixing schemes, this effect was even more pronounced, similar to the thermodynamic effect. Despite the complexity of aerosol-precipitation relationship, the suppression effect observed in our study can be attributed primarily to the aerosol-induced stability leading to weaker convection, as evidenced by the changes in the PBL height.

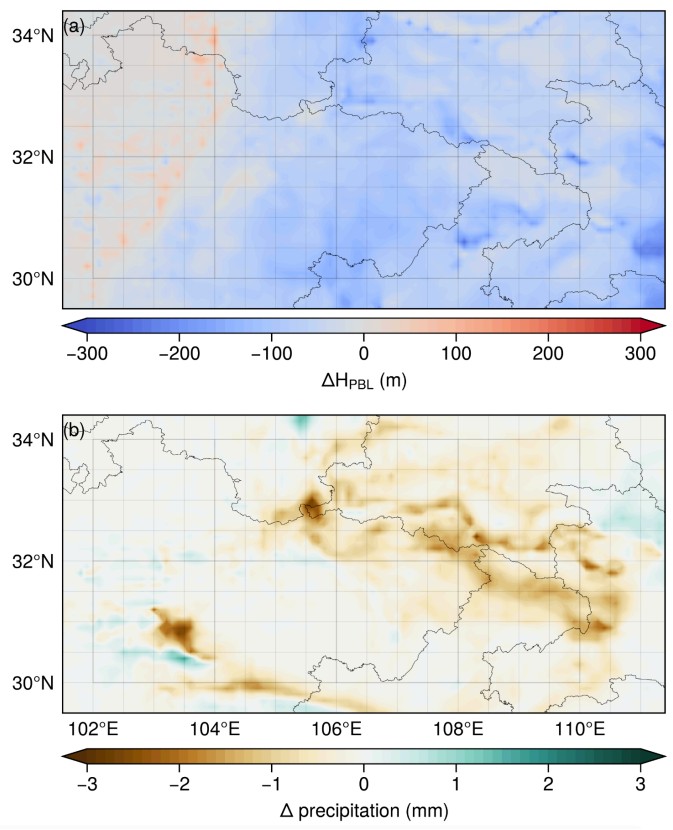

**Figure 11. Anomalies in (a) height of planetary boundary layer and (b) accumulative precipitation between the AI-NAOS module and the control scheme over a specific region within the precipitation center (101.5–112.5°E, 29.5–34.5°N).**

## 4 Conclusion

A new aerosol optical module, referred to as AI-NAOS, was developed to incorporate the nonsphericity and inhomogeneity of aerosol particles. Using integrated DNNs, aerosol optical properties were effectively calculated based on NSIH aerosol models. This newly developed aerosol optical module was on-line coupled with the chemical weather model, GRAPES_Meso5.1/CUACE.

In the AI-NAOS scheme, BC was assumed to be fractal aggregates while soil dust was modelled as super-spheroids. Both of these insoluble aerosols were either fully or partially encapsulated with hygroscopic aerosols, which were treated with the volume-mixing method. The aerosol optical properties were computed using the IITM algorithm. Two databases of optical properties for BC and SD were established, including more than 700, 000 records. Direct incorporating the two optical property databases into the aerosol parameterization scheme using the look-up table method is inconvenient. This method consumes a significant amount of storage and introduces additional errors from interpolation. To address this issue, two DNNs were trained based on the two optical property databases. A unified interface was implemented in the AI-NAOS scheme to calculate the optical properties of BC and SD using the DNNs.

Real-scenario simulations were conducted to assess the NSIH effect of BC aerosols using the GRAPES_Meso5.1/CUACE model. In the simulation, five aerosol treatments were applied, including the control experiment without aerosol optical effects, the AI-NAOS module, and three spherical schemes.

The AI-NAOS module provided a moderate estimation of AAOD, falling between the underestimation from the external-mixing scheme and the overestimation from the core-shell and the volume-mixing schemes. For example, the AAOD values based on the three spherical schemes were 70.4%, 125.3%, and 129.3% over Sichuan Basin, benchmarked to the AI-NAOS results. The absorption enhancements were 1.42, 1.79, and 1.86 based on the AI-NAOS, core-shell, and volume-mixing schemes. It was clear that the lowest absorption enhancement value was induced under the AI-NAOS module, indicating a favorable choice that matched low $E_{abs}$ measurements. It is worth noting that the asymmetry factor based on the NSIH scheme was the largest among all schemes, indicating the weakest backscattering fraction.

The aerosol shortwave DRE is directly influenced by the NSIH effect. Compared to the external-mixing scheme, the DRE induced by the NSIH effect was +1.6, +1.8, and +1.8 W/m$^2$ at TOA and -2.9, -5.3, and -4.1 W/m$^2$ at the surface over three regions, respectively. This effect was stronger within the core-shell and volume-mixing schemes, which could mainly be attributed to the difference in aerosol absorption. Compared to the AI-NAOS scheme, the DREs in the atmosphere were 80.7%, 112.5%, and 118.9% over the Sichuan Basin based on the three spherical schemes, respectively.

The NSIH effect also resulted in changes in the thermal structure. The heating rates within the AI-NAOS module were 0.64, 0.72, and 0.66 K/day on the ground and 0.29, 0.34, and 0.31 K/day at 700 hPa. Compared to the external-mixing scheme, the NSIH effect could enhance the heating rate, reaching at least 23%. This enhancement could be stronger within the core-shell and volume-mixing schemes, reaching 30% and 41%. The temperature anomalies under the AI-NAOS scheme were +0.17, +0.12, and +0.14 K at 700 hPa and -0.83, -1.11, and -1.37 K on the ground over three regions, respectively.

Compared to the external-mixing scheme, the NSIH effect could enhance the temperature anomalies by at least 21% at 700 hPa and 13% on the ground. This anomaly could be larger within the core-shell and volume-mixing schemes. The vertically averaged temperature anomalies of the three spherical schemes were 96%, 113%, and 108% benchmarked to the AI-NAOS scheme over the Sichuan Basin. As a result, the PBL becomes more stable, with its height decreasing by at least -11 m due to the NSIH effect. The precipitation was suppressed with weaker convection in the stable atmosphere, being -0.24 mm under

the AI-NAOS module. Compared to the external-mixing scheme, the NSIH effect enhanced the suppression effect by 15%.

        In this study, a comprehensive assessment was conducted to analyze the NSIH effect of BC aerosols. The findings indicated that the NSIH effect cannot be disregarded in weather forecast models. A crucial aspect is that the estimation of aerosol absorption can be improved by incorporating the NSIH effect. However, additional real-case studies, encompassing weather and climate scales, should be undertaken in the future.

485        Temporarily, the refractive indices and fractal dimension of BC were fixed. As the AI-NAOS module is driven by DNNs, it is convenient to adjust refractive indices without rebuilding the DNN. This flexibility proves advantageous for investigating the absorption ability of BC. Additionally, there are plans to train a new DNN that incorporates BC particles with various fractal dimensions, aiming to support more detailed research in this area. As for SD, the database of coated super-spheroid for large particles will be updated when a computational program based on the ray tracing technique is

available. In the current AI-NAOS module, the nonsphericity and inhomogeneity are only considered with insoluble aerosol particles. Although hygroscopic aerosols, such as sea salt, may be completely dissolved in aerosol water after deliquescence, a solid core coated with liquid saline still exists until the relative humidity reaches 97% (Zeng et al., 2013). The NSIH effect has a significant impact on the single-scattering properties of sea salt, especially with respect to depolarization (Bi et al., 2018b; Kanngießer and Kahnert, 2021; Lin and Bi, 2024). Additionally, it was found that the NSIH effect on optical

properties could induce a cooling effect, weaken vertical velocity and reduce the accumulated rainfall of Typhoon Fitow (Zhu et al., 2022). Note that the NSIH effect of sea salt aerosol is not important in this study, because we focused on the land area. In the future, we will consider the NSIH features of sea salt particles, developing a more comprehensive nonspherical and inhomogeneous aerosol optical scheme. Furthermore, it is anticipated that the AI-NAOS scheme will be routinely updated and implemented in various chemical weather, climate, and radiation transfer models, as well as utilized for data

assimilation purposes.

**Appendix A. Deep neural networks of bulk optical properties.**

        As shown in Table A1, there were three input parameters ($x$, $mr_{hygro}$, $mi_{hygro}$) for spheres in the volume-mixing scheme and six parameters ($x$, $mr_{hygro}$, $mi_{hygro}$, $vf$, $mr_{insol}$, $mi_{insol}$) for the core-shell model and two NSIH models of BC and SD. The architecture of the DNNs included fully connected (FC) layers and duplicated residual blocks, as depicted in Figure A1.

Specifically, the first two hidden layers consisted of fully connected layers with 40 and 20 nodes, respectively. The following setup included three branches corresponding to the extinction efficiency, single-scattering albedo, and asymmetry

factor. In each branch, two residual blocks were included and each block was composed by two fully-connected layers with 20 nodes and a shortcut connection. The architecture is similar to our previous work (Wang et al., 2023b). However, in this study, the DNNs were applied to bulk optical properties in various size bins, and the scattering matrices were excluded from the database. Additionally, we followed the method proposed in Wang et al. (2022b) to constrain the extrapolation of the DNN for large coated super-spheroids by using the asymptotic values of the core-shell model.

**Table A1. DNNs of bulk optical properties (used in the AI-NAOS module).**

| Aerosol | Particle Model | DNN input | DNN output |
|---|---|---|---|
| Hygroscopic aerosols | volume-mixing (sphere) | $(x, mr_{hygro}, mi_{hygro})$ | |
| BC | core-shell | | $(Q_{ext}, <SSA>, <G>)$ |
| | encapsulated fractal soot | $(x, mr_{hygro}, mi_{hygro}, vf, mr_{insol}, mi_{insol})$ | |
| SD | coated super-spheroid | | |

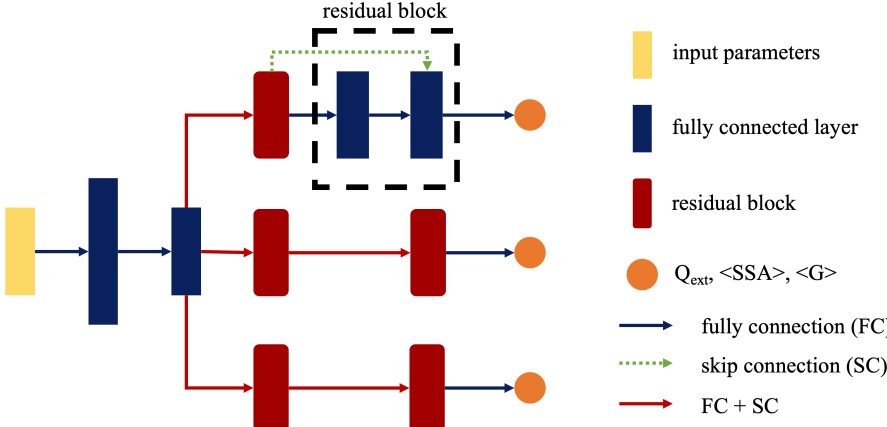

**Figure A1. The architecture of DNNs in this study.**

**Appendix B. Training of DNN and Tuning of Hyperparameters**

All DNNs used in this study were trained using the same architecture and configurations. Here, we specifically discuss the DNN for the encapsulated fractal model.

Firstly, the dataset of bulk optical properties was divided into three parts: 75% for training, 10% for validation, and 15% for testing. Since the dataset was well-organized, these three parts were randomly sampled without redistribution. Notably, the training dataset was not subjected to pre-processing such as normalization. We found that the DNN performed well even without the pre-processing step, making it more convenient for application as no additional data transforms were required before and after inference.

Next, a series of configurations were determined. Based on the same DNN architecture, the Leaky Rectified Linear Unit

with a negative slope of 0.01 was chosen as the activation function. The loss value was calculated using RMSE, and the parameters of DNN was optimized using the Adam algorithm. Several hyperparameters, including batch size, initial learning rate, and number of nodes in the first fully connected (FC) layer, were fine-tuned. The learning rate was annealed using a cosine function and the number of nodes in subsequent hidden layers was set to be half of the number in the first FC layer. To determine the optimal hyperparameters, the Asynchronous Successive Halving Algorithm (ASHA) was employed. The

search process was allowed to proceed for a maximum of 200 epochs, unless early stopping criteria were met. The results of the hyperparameter tuning are summarized in Table A2.

It was clear from the results that the loss value decreased as the number of nodes increased, but this improvement became less significant when the number of nodes exceeded 40. After balancing DNN performance and efficiency, the number of nodes in the first FC layer, batch size, and initial learning rate were set to the values of 40, 200, and 5×10-3,

respectively.

**Table A2. Optimal values of hyperparameters**

| Nodes of first FC layer | Batch size | Initial learning rate | Loss value in validation |
|---|---|---|---|
| 60 | 100 | $10^{-3}$ | $2.3 \times 10^{-3}$ |
| 50 | 100 | $10^{-3}$ | $2.6 \times 10^{-3}$ |
| 40 | 200 | $5 \times 10^{-3}$ | $3.4 \times 10^{-3}$ |
| 30 | 100 | $5 \times 10^{-3}$ | $8.8 \times 10^{-3}$ |
| 20 | 200 | $5 \times 10^{-3}$ | $1.2 \times 10^{-2}$ |

**Appendix C. Generalizability**

To assess the generalizability of the DNN, we compared its predictions against the rigorous results obtained from the

IITM for microphysical parameters not included in the database. Given that the parameter range in the database encompasses nearly all potential values of refractive indices and volume fractions, our focus was primarily on evaluating the interpolation accuracy. Specifically, we chose complex refractive indices of 1.33+0i for hygroscopic aerosols (coat) and 1.95+0.79i for black carbon (core), with a volume fraction set at 0.33. However, we examined the performance of both interpolation and extrapolation in terms of the size parameter, which extended beyond the database maximum value of 16 to 24. We did not

consider size parameters larger than 24, due to the significant increase in computational time required by the IITM and the rarity of particles exceeding these values in reality. As illustrated in Figure A2, the RMSE values of bulk extinction efficiency, single-scattering albedo, and asymmetry factor were 0.0149, 0.0065, and 0.0050, respectively. At the size parameter of 24, the errors of these three bulk optical properties were +0.012, -0.004, +0.001, respectively. Overall, the DNN demonstrated good generalizability. It is worth noting that the size parameter in this study was defined for the midpoint of

the size bin. For a more detailed analysis of the DNN's generalizability for the single-scattering properties without size integration, please refer to Wang et al. (2023b).

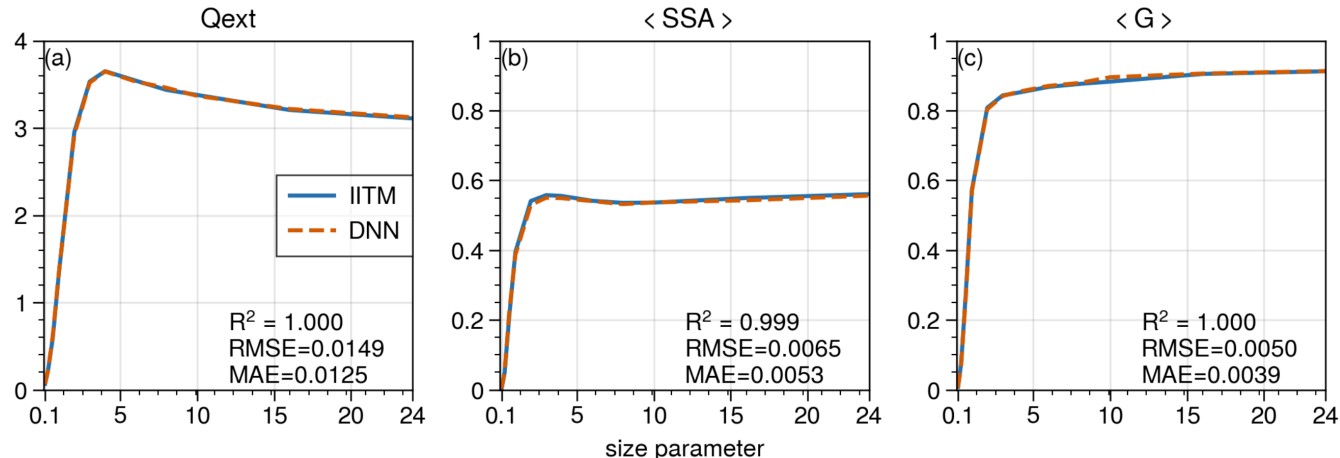

**Figure A2. Comparison of bulk optical properties of encapsulated fractal aggregates computed from the IITM and DNN: (a) extinction efficiency, (b) single-scattering albedo, and (c) asymmetry factor. The complex refractive indices of hygroscopic aerosols**
**and BC were 1.33+0i and 1.95+0.79i, respectively. The volume fraction was 0.33.**

## Appendix D. Time Complexity

Once the DNN is well trained from the IITM database, the DNN is significantly faster than the IITM for obtaining new results. An experiment was carried out on a dual-CPU node equipped with 28 processors (Intel Xeon E5-2680 v4). As a parallel algorithm, the IITM leveraged all 28 threads to compute the optical properties of a single particle. Subsequently,
bulk optical properties were obtained based on 10 quadrature points. In contrast, the DNN was employed to directly calculate the inferred values within a single thread over a million iterations. The results, illustrated in Figure A3, demonstrate a clear trend: the computational cost of IITM increases sharply with the size parameter, whereas the inference time of DNN remains relatively stable. Compared to the IITM, the computational efficiency of the DNN was found to be $10^9$, $10^{10}$, $10^{11}$, and $10^{12}$ times higher for size parameters of 2.0, 3.0, 8.0, and 16.0, respectively. Notably, the DNN has been verified as a reliable
acceleration algorithm, even for the efficient Lorenz–Mie theory, achieving a speedup of $10^3$ times (Kumar et al., 2024).

In weather chemical models, the aerosol optical properties were commonly accessed using look-up tables. However, directly comparing the AI-NAOS scheme with the look-up table method was challenging due to the lack of integration of the bulk optical property database into weather chemical models. Therefore, we evaluated AI-NAOS against other existing schemes, including the external-mixing scheme in GRAPES_Meso5.1/CUACE, core-shell and volume-mixing schemes in
WRF-Chem V4.2.1. Additionally, we introduced a modified version of AI-NAOS scheme, denoted as AI-NAOS*, which excluded DNN inference for zero AOD case, allowing for the separation of prior processes for internal mixing and the DNN inference. We conducted triplicate 12-hour simulations with 75,551 grids using 4 nodes, with a grid spacing of 0.1° for GRAPES/CUACE and 9 km for WRF-Chem. The total computation times for these simulations are summarized in Figure

A3. Compared to core-shell and volume-mixing schemes, the AI-NAOS required an additional 2.2% and 2.8% of computational time, respectively. In contrast, when compared to the external-mixing scheme, the AI-NAOS needed extra 17.7% of time, which was comprised of 6.2% for the DNN inference and 11.4% for prior processes. Generally, there was not substantial difference between invoking the DNN and using look-up tables.

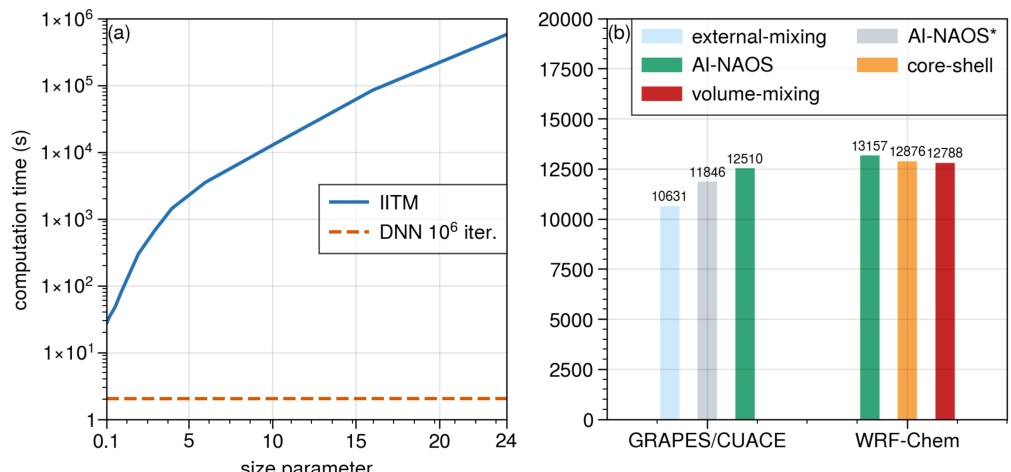

**Figure A3. (a) Comparison of computation time required by the IITM and DNN. (b) Comparison of computation time of simulations between external-mixing, AI-NAOS*, AI-NAOS, core-shell, and volume-mixing.**

*Data availability.* The data presented in this paper are available on Zenodo (https://doi.org/10.5281/zenodo.11183085; Wang et al., 2024a).

*Code availability.* The AI-NAOS aerosol optical module codes are available on Zenodo (https://doi.org/10.5281/zenodo.11181275; Wang et al., 2024b). The repository for GRAPES_Meso5.1/CUACE developed by Wang et al., 2022a, is in Zenodo (https://zenodo.org/records/7075751).

*Author contributions.* XW performed the model development and simulations. HW and YW contributed to the CUACE model. WH and XS contributed to the GRAPES model. LB and XZ supervised this project. All authors contributed to the writing or editing of the paper.

*Acknowledgement.* We used the computing facilities of the China HPC Cloud Computing Center. This work was supported by the National Key Scientific and Technological Infrastructure project Earth System Numerical Simulation Facility (EarthLab).

*Financial support.* This research was supported by the NSFC Major Project (42090030).

*Competing interests.* The contact author has declared that none of the authors has any competing interests.

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
