# Peer review of "AI-NAOS: An AI-Based Nonspherical Aerosol Optical Scheme for Chemical Weather Model GRAPES Meso5.1/CUACE"

_Geoscientific Model Development, 2024_

## Author Comment (AC3)

**RC1: 'Comment on gmd-2024-51', Anonymous Referee #1, 12 Jul 2024**

This manuscript introduces a deep learning approach to estimate the optical properties of internally mixed aerosol particles, considering the non-sphericity of insoluble particles. The research has significant implications for atmospheric models.

**Response: We appreciate the reviewer's positive feedback and constructive suggestions for improving the manuscript.**

**Comments for Improvement:**

1. Generalizability: While the discussed machine learning (ML) approach demonstrates impressive accuracy, concerns remain about its generalizability. ML-based algorithms often struggle with generalization beyond their training data (Kumar et al., 2024). The authors should address this issue explicitly.

**Response: Thanks for your comment. Appendix C (Line 538) was included in the revised manuscript with more discussions about the DNN generalizability.**

**Appendix C. The DNN Generalizability**

**To assess the generalizability of the DNN, we compared its predictions against the rigorous results obtained from the IITM for microphysical parameters not included in the database. Given that the parameter range in the database encompasses nearly all potential values of refractive indices and volume fractions, our focus was primarily on evaluating the interpolation accuracy. Specifically, we chose complex refractive indices of 1.33+0i for hygroscopic aerosols (coat) and 1.95+0.79i for black carbon (core), with a volume fraction set at 0.33. However, we examined the performance of both interpolation and extrapolation in terms of the size parameter, which extended beyond the database maximum value of 16 to 24. We did not consider size parameters larger than 24, due to the significant increase in computational time required by the IITM and the rarity of particles exceeding these values in reality. As illustrated in Figure A2, the RMSE values of bulk extinction coefficient, single-scattering albedo, and asymmetry factor were 0.0149, 0.0065, and 0.0050, respectively. At the size parameter of 24, the errors of these three bulk optical properties were +0.012, -0.004, +0.001, respectively. Overall, the DNN demonstrated good generalizability. It is worth noting that the size parameter in this study was defined for the midpoint of the size bin. For a more detailed analysis of the DNN's generalizability for the single-scattering properties without size integration, please refer to Wang et al. (2023b).**

[Figure]

**Figure A2.** Comparison of bulk optical properties of encapsulated fractal aggregates computed from the IITM and DNN: (a) extinction coefficient, (b) single-scattering albedo, and (c) asymmetry factor. The complex refractive indices of hygroscopic aerosols and BC were 1.33+0i and 1.95+0.79i, respectively. The volume fraction was 0.33.

2. Comparison with Observed AOD: Including a comparison of results with observed Aerosol Optical Depth (AOD) would enhance the manuscript. This validation step provides valuable context.

**Response: Thanks for your valuable suggestion. We have incorporated a comparison between MODIS AOD product and our model simulations into the manuscript, specifically in Line 313.**

**As shown in Figure 6, we have used the daily AOD product from the Moderate Resolution Imaging Spectroradiometer (MODIS) to validate our simulations on January 13. The spatial distribution of AOD observed by MODIS exhibits a similar pattern to our simulations, with high AOD values detected over three regions characterized by high anthropogenic emissions: the Sichuan Basin, the Middle Yangtze Plain, and the North China Plain.**

**To gain a better understanding of AOD distribution pattern, we calculated the probability distribution function (PDF) over a wide region where high AOD values were observed (105–118°E, 27–40°N). Due to the presence of missing values in the MODIS AOD product, corresponding values in our simulations were also omitted. The simulations reveal a more concentrated distribution pattern, with the highest AOD values being slightly lower than those observed by MODIS. The median values of simulated AOD within external-mixing, AI-NAOS, core-shell, and volume-mixing schemes are 0.206, 0.225, 0.232, and 0.238, respectively. The MODIS median value of 0.217 falls between the external-mixing and AI-NAOS schemes, indicating a good agreement between our simulations and the observations.**
**This validation step provides valuable context and enhances the robustness of our findings.**

[Figure]

**Figure 6. Spatial distribution of daily AOD on January 13 from (a) MODIS product [MOD08_D3] and (b) simulations with AI-NAOS. (c) AOD probability distribution function of the MODIS product and simulations with external-mixing, AI-NAOS, core-shell, and volume-mixing schemes.**

3. Time Complexity: The authors do not discuss the time complexity of the developed approach. One of the main reasons scientists are increasingly pursuing ML is due to its computational cost advantages.

**Response: Regarding the time complexity involved in the current ML study, we would like to highlight the following points:**

**(1) Once the DNN is well trained from the IITM database, the DNN is significantly faster than the IITM for obtaining new results. The DNN demonstrates excellent generalizability, ensuring prediction accuracy, albeit slightly lower than the IITM results. This aligns with the main reason of increasingly pursuing ML in a broad scientific community.**

**(2) In the CUACE model, there should not be a significant difference between calling the DNN and using look-up-table, as the look-up-table method is generally quite efficient. The primary advantage of the DNN lies in its high flexibility and accuracy rather than its speed. Appendix D (Line 556) was included in the revised manuscript with more discussions about the time complexity.**

**Appendix D. Time Complexity**
**Once the DNN is well trained from the IITM database, the DNN is significantly faster than the IITM for obtaining new results. An experiment was carried out on a dual-CPU node equipped with 28 processors (Intel Xeon E5-2680 v4). As a parallel algorithm, the IITM leveraged all 28 threads to compute the optical properties of a single particle. Subsequently, bulk optical properties were obtained based on 10 quadrature points. In contrast, the DNN was employed to directly calculate the inferred values within a single thread over a million iterations. The results, illustrated in Figure A3, demonstrate a clear trend: the computational cost of IITM increases sharply with the size parameter, whereas the inference time of DNN remains relatively stable. Compared to the IITM, the computational efficiency of the DNN was found to be $10^9$, $10^{10}$, $10^{11}$, and $10^{12}$ times higher for size parameters of 2.0, 3.0, 8.0, and**

16.0, respectively. Notably, the DNN has been verified as a reliable acceleration algorithm, even for the efficient Lorenz–Mie theory, achieving a speedup of $10^3$ times (Kumar et al., 2024).

In weather chemical models, the aerosol optical properties were commonly accessed using look-up tables. However, directly comparing the AI-NAOS scheme with the look-up table method was challenging due to the lack of integration of the bulk optical property database into weather chemical models. Therefore, we evaluated AI-NAOS against other existing schemes, including the external-mixing scheme in GRAPES_Meso5.1/CUACE, core-shell and volume-mixing schemes in WRF-Chem V4.2.1. Additionally, we introduced a modified version of AI-NAOS scheme, denoted as AI-NAOS*, which excluded DNN inference for zero AOD case, allowing for the separation of prior processes for internal mixing and the DNN inference. We conducted triplicate 12-hour simulations with 75,551 grids using 4 nodes, with a grid spacing of 0.1° for GRAPES/CUACE and 9 km for WRF-Chem. The total computation times for these simulations are summarized in Figure A3. Compared to core-shell and volume-mixing schemes, the AI-NAOS required an additional 2.2% and 2.8% of computational time, respectively. In contrast, when compared to the external-mixing scheme, the AI-NAOS needed extra 17.7% of time, which was comprised of 6.2% for the DNN inference and 11.4% for prior processes. Generally, there was not substantial difference between invoking the DNN and using look-up tables.

[Figure]

Figure A3. (a) Comparison of computation time required by the IITM and DNN. (b) Comparison of computation time for simulations, showcasing the differences between external-mixing, AI-NAOS, core-shell, and volume-mixing schemes.

4. Figure 3: Clarify whether the results in Figure 3 pertain to the test or training dataset. Additionally, provide details on pre-processing and post-processing steps, which are currently missing.

**Response: Thank you for pointing out the missing information. To clarify, the data points presented were randomly sampled from the entire dataset, which includes training, validation, and testing sets. Actually, the data were not subjected to any pre-processing steps. For further reference, please refer to Appendix B, which is also mentioned in our response to Comment 5.**

5. Tuning of Hyperparameters: While the network's results are extraordinary, the manuscript lacks information on hyperparameters and their tuning. Including this would improve transparency.

**Response: Thanks for your suggestion. We have included Appendix B to elaborate on the DNN training and hyperparameter tuning process (Line 516).**

**Appendix B. Training of DNN and Tuning of Hyperparameters**

**All DNNs used in this study were trained using the same architecture and configurations. Here, we specifically discuss the DNN for the encapsulated fractal model.**

**Firstly, the dataset of bulk optical properties was divided into three parts: 75% for training, 10% for validation, and 15% for testing. Since the dataset was well-organized, these three parts were randomly sampled without redistribution. Notably, the training dataset was not subjected to pre-processing such as normalization. We found that the DNN performed well even without the pre-processing step, making it more convenient for application as no additional data transforms were required before and after inference.**

**Next, a series of configurations were determined. Based on the same DNN architecture, the Leaky Rectified Linear Unit with a negative slope of 0.01 was chosen as the activation function. The loss value was calculated using RMSE, and the parameters of DNN was optimized using the Adam algorithm. Several hyperparameters, including batch size, initial learning rate, and number of nodes in the first fully connected (FC) layer, were fine-tuned. The learning rate was annealed using a cosine function and the number of nodes in subsequent hidden layers was set to be half of the number in the first FC layer. To determine the optimal hyperparameters, the Asynchronous Successive Halving Algorithm (ASHA) was employed. The search process was allowed to proceed for a maximum of 200 epochs, unless early stopping criteria were met. The results of the hyperparameter tuning are summarized in Table A2.**

**It was clear from the results that the loss value decreased as the number of nodes increased, but this improvement became less significant when the number of nodes exceeded 40. After balancing DNN performance and efficiency, the number of nodes in the first FC layer, batch size, and initial learning rate were set to the values of 40, 200, and $5\times10^{-3}$, respectively.**

**Table A2. Optimal values of hyperparameters**

| Nodes of first FC layer | Batch size | Initial learning rate | Loss value in validation |
|:---:|:---:|:---:|:---:|
| 60 | 100 | $10^{-3}$ | $2.3 \times 10^{-3}$ |
| 50 | 100 | $10^{-3}$ | $2.6 \times 10^{-3}$ |
| 40 | 200 | $5 \times 10^{-3}$ | $3.4 \times 10^{-3}$ |
| 30 | 100 | $5 \times 10^{-3}$ | $8.8 \times 10^{-3}$ |
| 20 | 200 | $5 \times 10^{-3}$ | $1.2 \times 10^{-2}$ |

6. Methodology Reorganization: Consider reorganizing the methodology section. Present results from AI-NAOS (lines 174-190, including Figure 2) after describing the approach in detail.
**Response: Thanks for your suggestion. This part was moved to Line 212.**

7. Move lines 208-216 and Figure 3 to the results section for better flow.
**Response: Thanks for your suggestion. This part was moved to the result section as a new subsection titled "3.1 Performance of Deep Neural Networks" (Line 259). Furthermore, the "model configuration" subsection was moved to method section (Line 232).**

**The following are additional revisions:**

**1. Figure 1 were displayed in another viewing angle for better understanding.**

[Figure]

**Figure 1. Optical modelling for (a) fractal aggregates framework of black carbon (BC) partially encapsulated with spherical coating of hygroscopic aerosols and (b) super-spheroid framework of soil dust (SD) fully coated with another super-spheroid of hygroscopic aerosols with various volume fractions.**

**2. A figure was added to illustrate the framework of AI-NAOS module (Figure 2).**

[Figure]

**Figure 2. The framework of AI-NAOS module.**

**3. Figure 10 was modified. Pressure levels ranged from 700 hPa to the surface instead of 1000 hPa in the new version and vertical averaged values were corrected. The main conclusion, "The NSIH effect could enhance the short-wave heating rate, reaching 20%", was changed to "The NSIH effect could enhance the short-wave heating rate, reaching 23%".**

[Figure]

**Figure 10. Vertical profiles of (a-c) temperature and (d-f) short-wave heating rate anomalies based on four aerosol optical schemes (external-mixing, AI-NAOS, core-shell, and volume-mixing). The solid lines represent the median value and the shaded areas encompass the range from the 25 to 75 percentage.**

**4. The colour schemes were modified in Figure 3, 7, 8, 10, for the convenience of readers with colour vision deficiencies.**

**5. The repository for GRAPES_Meso5.1/CUACE was added (Line 585)**
**The repository for GRAPES_Meso5.1/CUACE developed by Wang et al., 2022a, is in Zenodo (https://zenodo.org/records/7075751).**

---

## Author Comment (AC4)

**RC2: 'Comment on gmd-2024-51', Anonymous Referee #2, 21 Oct 2024**

The manuscript introduces AI-NAOS, an AI-based module that incorporates nonspherical and inhomogeneous aerosol particles into radiative transfer simulations. The use of deep learning and advanced optical modeling is an innovative approach in this field, significantly improving accuracy in predicting the direct radiative effects of aerosols. Real-case simulations provide substantial evidence of the AI-NAOS module's impact on atmospheric thermodynamic structures and precipitation patterns, enhancing the understanding of aerosols in weather modeling. In general, the manuscript is well-organized. Only minor revision is needed before publication:

**Response: We appreciate the reviewer's positive comment on this study and constructive suggestions for improving the manuscript.**

The model ran for 72 hours starting from January 12, 2018, with only 24 hours of spin-up time, which seems rather short. Is it possible to extend the model's spin-up time and total run time?

**Response: Thanks for your suggestion. Currently, we are integrating the module into the CESM2/CAM6 climate model, in which case we will increase the run time for tests as needed. For the weather-scale case studied here, while the spin-up and total run time can be extended for additional analysis, we believe that the chosen spin-up and total run time of 24 hours and 72 hours, respectively, are adequate for our purposes.**

**It is worth noting that a spin-up time of 12 hours is commonly used in mesoscale numerical weather prediction models (Dzebre et al., 2019; Wang et al., 2021). For certain initial conditions with disturbing weather events, a longer spin-up time may be required. In our case with no disturbing weather events, the 24-hour spin-up was sufficient for the model to reach a physical equilibrium state (Liu et al., 2023).**

**Regarding the total run time, studies have indicated that a duration ranging from 1 day to 5 days is acceptable in a weather-scale case study(Di et al., 2015; Iguchi et al., 2012). Given the current configuration of our model and the short time available for manuscript revision, we have not extended the run time for further analysis.**

Is 'deep neural network (DNN)' used in the text as a general term for a class of methods, or does it refer to a specific neural network method? Please clarify this further in the text.

**Response: Thank you for raising this question for clarification. DNN is used in the text as a general term referring to a series of neural networks with multiple hidden layers. However, in the context of this study, we specifically employ a multiple-target DNN that was developed in Wang et al. (2023). To provide further clarification, we have updated the manuscript with following inclusions (Line 181):**

**A multiple-target DNN model that has been designed to infer the single-scattering properties of encapsulated fractal aggregates of BC was adapted to bulk optical properties inference in this study.**

Why were different methods chosen for shortwave radiation and longwave radiation? I believe that RRTM can also be used for shortwave calculations. What considerations influenced this decision?

**Response: The use of different methods for shortwave and longwave radiation was based on the specific configurations and developments within the GRAPES_Meso5.1/CUACE model (Peng et al., 2022). We decided to maintain this configuration for consistency and simplicity. We understand that there are alternative approaches and configuration. We appreciate your insights for future considerations.**

Could you compare it with the latest research on non-spherical black carbon, such as the study by Chen, G., Liu, C., Wang, J., Yin, Y., & Wang, Y., JGRA, 2024, which accounts for the mixing state, nonsphericity, and heterogeneity effects of black carbon in its optical property parameterization within a climate model?

**Response: Thanks for the reviewer's comment. Chen et al. in their study published in JGRA have indeed made significant progress by establishing a 3-D dataset of bulk optical properties of non-spherical black carbon and integrating it into the climate model CAM6 using a look-up table method. Their work effectively addresses the overestimation of BC absorption by considering particle nonsphericity.**

**The primary difference between Chen's work and our NAOS framework are twofold:**

**Firstly, the database employed differ significantly. In our research, we use the IITM algorithm to create our database, which offers a more efficient and accurate alternative to the discrete dipole approximation (DDA) method used by Chen et al. Thus, our database is larger and more comprehensive. For example, Chen's 3-D database includes only a single value of complex refractive index (RI) of BC and consider just two values for the real part and two for the imaginary part of RI for coating materials. Additionally, the size range of BC particles is limited, leading to the use of spherical core-shell models for large bins in their dataset. This limitation means that the nonsphericity of large BC particles in accumulation mode and all BC particles in primary carbon mode have not been accounted for.**

**Secondly, the look-up table method necessitates interpolation, which may introduce non-negligible numerical errors, especially when interpolating points are sparse. In contrast, the DNN approach provides continuous predictions, outperforming traditional interpolation. While both studies apply similar encapsulated fractal aggregates models, Chen's work includes a more diverse range of shapes. Similarly, we plan to incorporate additional shapes with varying fractal dimensions into our NAOS framework in future updates.**

I believe there is still considerable uncertainty regarding the impact on precipitation in this section. It's worth further consideration whether to include this in the article. Additionally, while not absolutely necessary, it might be beneficial to examine changes in the boundary layer as well.

**Response: Thanks for your valuable feedback. We acknowledge the complexity of the relationship between DRE and precipitation, particularly in extreme weather scenarios. In our study, which focused on light winter rainfall, the suppression effect on precipitation can primarily be attributed to reduced convection due to decreased boundary layer height.**

To further elucidate this mechanism, we have conducted a detailed examination of boundary layer height and incorporated a new subfigure in Figure 11. Additionally, Section 3.5 has been revised accordingly:

Figure 11 depicts the anomalies in height of PBL and accumulated precipitation over the last 48 hours, comparing the AI-NAOS scheme with a control scheme of zero AOD. The analysis focused on a specific region (101.5-112.5°E, 29.5-34.5°N), encompassing parts of the Sichuan Basin and the Middle Yangtze Plain. The results indicate a suppression effect on precipitation accompanied by a decrease in PBL height in this region. The spatially averaged precipitation anomaly was -0.24 mm for the AI-NAOS module, with anomalies of -0.21, -0.26, and -0.27 mm for the external-mixing, core-shell, and volume-mixing schemes, respectively. The PBL height anomalies for AI-NAOS and the three spherical schemes were -56.8, -48.7, -63.0, and -64.4 meters, respectively. Notably, the suppression effect was less significant in the external-mixing scheme. The NSIH effect amplified the suppression effect by approximately 15%. In the core-shell and volume-mixing schemes, this effect was even more pronounced, similar to the thermodynamic effect. Despite the complexity of aerosol-precipitation relationship, the suppression effect observed in our study can be attributed primarily to the aerosol-induced stability leading to weaker convection, as evidenced by the changes in the PBL height.

[Figure]

Figure 11. Anomalies in (a) height of planetary boundary layer and (b) accumulative precipitation between the AI-NAOS module and the control scheme over a specific region within the precipitation center (101.5–112.5°E, 29.5–34.5°N).

Some minor comments:

In Line 40, omega is used to represent SSA, while in Line 271, <SSA> is used. Please ensure consistency. The same applies to the asymmetry factor.

**Response: The omega symbol was used to represent SSA of single particle while <SSA> was used to represent bulk SSA. These two variables were represented by two different symbols and the relationship was described in Eq. 7.**

$$< SSA > = \frac{\int_{D_{min}}^{D_{max}} \omega q_{ext} D^2 n(D) dD}{\int_{D_{min}}^{D_{max}} q_{ext} D^2 n(D) dD} , \tag{7}$$

Line 80: It is usually referred to as 'WRF-Chem' rather than 'WRF/chem'.
**Response: Corrected. Thanks.**

The vertical axis of Figure 6b is labeled 'atmo. abs.', which seems less rigorous than 'DRE'.

**Response: Thanks! After some considerations, we decided to use ''Atmo. Abs.' for better understanding since the difference of net solar flux between TOA and surface could be totally attributed to absorption rather than scattering.**

Section 3.4 appears twice; the precipitation part should be labeled as 3.5.

**Response: Corrected. Thanks!**

**References:**

Di, Z., Duan, Q., Gong, W., Wang, C., Gan, Y., Quan, J., Li, J., Miao, C., Ye, A., and Tong, C.: Assessing WRF model parameter sensitivity: A case study with 5 day summer precipitation forecasting in the Greater Beijing Area, Geophysical Research Letters, 42, 579–587, https://doi.org/10.1002/2014GL061623, 2015.

Dzebre, D. E. K., Acheampong, A. A., Ampofo, J., and Adaramola, M. S.: A sensitivity study of Surface Wind simulations over Coastal Ghana to selected Time Control and Nudging options in the Weather Research and Forecasting Model, Heliyon, 5, e01385, https://doi.org/10.1016/j.heliyon.2019.e01385, 2019.

Iguchi, T., Matsui, T., Tokay, A., Kollias, P., and Tao, W.-K.: Two distinct modes in one-day rainfall event during MC3E field campaign: Analyses of disdrometer observations and WRF-SBM simulation, Geophysical Research Letters, 39, https://doi.org/10.1029/2012GL053329, 2012.

Liu, Y., Zhuo, L., and Han, D.: Developing spin-up time framework for WRF extreme precipitation simulations, Journal of Hydrology, 620, 129443, https://doi.org/10.1016/j.jhydrol.2023.129443, 2023.

Peng, Y., Wang, H., Zhang, X., Zheng, Y., Zhang, X., Zhang, W., Liu, Z., Gui, K., Liu, H., Wang, Y., and Che, H.: Aerosol-radiation interaction in the operational atmospheric chemistry model GRAPES_Meso5.1/CUACE and its impacts on mesoscale NWP in Beijing-Tianjin-Hebei, China,

Atmospheric Research, 280, 106402, https://doi.org/10.1016/j.atmosres.2022.106402, 2022.

Wang, X., Tolksdorf, V., Otto, M., and Scherer, D.: WRF-based dynamical downscaling of ERA5 reanalysis data for High Mountain Asia: Towards a new version of the High Asia Refined analysis, International Journal of Climatology, 41, 743–762, https://doi.org/10.1002/joc.6686, 2021.

**The following are additional revisions:**

**1. Figure 1 were displayed in another viewing angle for better understanding.**

[Figure]

**Figure 1. Optical modelling for (a) fractal aggregates framework of black carbon (BC) partially encapsulated with spherical coating of hygroscopic aerosols and (b) super-spheroid framework of soil dust (SD) fully coated with another super-spheroid of hygroscopic aerosols with various volume fractions.**

**2. A figure was added to illustrate the framework of AI-NAOS module (Figure 2).**

[Figure]

**Figure 2. The framework of AI-NAOS module.**

**3. Figure 10 was modified. Pressure levels ranged from 700 hPa to the surface instead of 1000 hPa in the new version and vertical averaged values were corrected. The main conclusion, "The NSIH effect could enhance the short-wave heating rate, reaching 20%", was changed to "The NSIH effect could enhance the short-wave heating rate, reaching 23%".**

[Figure]

**Figure 10. Vertical profiles of (a-c) temperature and (d-f) short-wave heating rate anomalies based on four aerosol optical schemes (external-mixing, AI-NAOS, core-shell, and volume-mixing). The solid lines represent the median value and the shaded areas encompass the range from the 25 to 75 percentage.**

**4. The color schemes were modified in Figure 3, 7, 8, 10, for the convenience of readers with color vision deficiencies.**

**5. The repository for GRAPES_Meso5.1/CUACE was added (Line 585)**
**The repository for GRAPES_Meso5.1/CUACE developed by Wang et al., 2022a, is in Zenodo (https://zenodo.org/records/7075751).**